# Hydrocarbon Compatible SOFC Anode Catalysts and Their Syntheses: A Review

**Selvaraj Senthil Kumar and Singanahally ThippaReddy Aruna ***

Surface Engineering Division, Council of Scientific and Industrial Research-National Aerospace Laboratories, Bangalore 560017, India; ssenthil@nal.res.in
* Correspondence: aruna_reddy@nal.res.in

**Abstract:** With the fast depleting rate of fossil fuels, the whole world is looking for promising energy sources for the future, and fuel cells are perceived as futuristic energy sources. Out of the different varieties of fuel cells, solid oxide fuel cells (SOFCs) are promising due to their unique multi-fuel operating capability without the need for an external reformer. Nonetheless, the state-of-the-art anode material Ni–YSZ undergoes carburization in presence of hydrocarbons (HCs), resulting in performance degradation. Several strategies have been explored by researchers to overcome the issue of carburization of the anode. The important strategies include reducing SOFC operating temperature, adjustment of steam: carbon ratio, and use of alternate anode catalysts. Among these, the use of alternate anodes is a promising strategy. Apart from the carburization issue, the anode can also undergo sulfur poisoning. The present review discusses carburization and sulfur poisoning issues and the different strategies that can be adopted for tackling them. The quintessence of this review is to provide greater insight into the various developments in hydrocarbon compatible anode catalysts and into the synthesis routes employed for the synthesis of hydrocarbon compatible anodes.

**Keywords:** SOFC; hydrocarbons; internal reforming; synthesis; carburization; sulfur poisoning; catalyst





## 1. Introduction

Fossil fuels such as gasoline, coal, and jet fuels are not renewable. Burning these limited fuel resources not only increases air pollution but also leads to a severe economic crisis. However, producing power from renewable sources still remains a challenge. Several eco-friendly power sources such as solar, wind, hydroelectric, and geothermal power sources can be used only in particular environments [1]. In the present scenario, batteries play a prominent role in portable devices. They are energy storage devices that have limited lifetimes and need to be disposed of in hazardous-waste landfills. In contrast, fuel cells are energy conversion devices that can exhibit near-zero emissions, are silent and effectual, and can operate in any environment [2,3].

Fuel cells are electrochemical devices that directly convert chemical energy into electrical energy, and fuel utilization efficiency of up to 85% can be achieved. Hence, it is possible to double the efficiency of power systems. Further, fuel cells can run in regenerative mode, i.e., they can also convert the excess electrical energy back to chemical energy. This feature enables fuel cells to couple with modern renewable technologies such as solar, wind, etc., to produce uninterrupted power. Thus, various types of fuel cell technologies are being pursued across the world. Among them, proton exchange membrane fuel cells (PEMFCs) and solid oxide fuel cells (SOFCs) are considered propitious technologies owing to their relatively higher efficiency. Further, the relatively striking level of development and potential for commercialization make these technologies the best alternatives to solve the future energy crisis. Thus, it is of paramount importance to adopt these fuel cells on a large scale for civil and military purposes [4,5].

PEMFC are low-temperature fuel cells that function in the temperature range of 70–90 °C, whereas SOFC operates at 800–1000 °C. The intermediate temperature (550–750 °C)

SOFC (IT-SOFC) is still at the nascent stages of development. However, high-temperature SOFCs have several fundamental advantages over low-temperature fuel cells. These advantages include high power density and fuel tolerance. Further, it is worth mentioning that other fuel cells such as PEMFCs require controlled hydration of electrolyte membrane, and there is an obvious difficulty in maintaining the hydration in colder environments. This problem does not persist in SOFCs as they operate at relatively higher temperatures. Moreover, the heat from spent steam during SOFC operation can be utilized in other process requirements. As the name indicates, SOFC is made of all solids construction, and it operates at high temperatures and generates clean, efficient power from easy-to-transport fuels in lieu of pure hydrogen. Due to their low sensitivity to fossil fuels and their tolerance of impurities, SOFCs are extremely suitable for the use of HCs for auxiliary power units (APU) for vehicles as well as for stationary applications. Thus, SOFCs can find application in all types of environments including harsh environments encountered by aircraft, submarines, etc. [6–9]. Thus, the high efficiency and its ability to handle hydrocarbon (HC) fuels (including biofuels) have made the HC fuel-based SOFC one of the possible solutions to future energy needs.

There are seminal reviews on SOFCs, and readers can discern more information on SOFCs from them [10,11]. The contemporary SOFC single cells are fabricated in a planar design, and each cell consists of (a) dense 8 mol% yttria stabilized cubic zirconia (YSZ) as the electrolyte, (b) porous strontium doped lanthanum manganite (LSM) as the cathode, and (c) porous Ni–YSZ as the anode. Single cells are stacked using stainless-steel interconnections with channels for gas flow. The SOFCs are mostly designed either in anode- or electrolyte-supported configurations, and they are referred to as anode-supported cells (ASC) and electrolyte-supported cells (ESC) (Figure 1). The steps involved in the fabrication of ASC and ESC are shown in the flowchart (Figure 2). The tapecasting and screen printing are the most commercially viable techniques universally used for the fabrication of SOFC single cells. For ASC, the anode tapes containing NiO–YSZ and pore formers are stacked; over that, a thin YSZ tape is co-cast or a separate tape is placed over the NiO–YSZ, pressed, sintered, and followed by screen printing of cathode paste on the electrolyte and sintered again at a lower temperature than the anode sintering temperature. In the case of ESC, electrolyte YSZ or scandia-stabilized zirconia (ScSZ) tapes are stacked and sintered, followed by the screen printing of the anode and its sintering, followed by screen-printing of the cathode and, finally, sintering it again.

Though SOFC is capable of producing electricity using HCs as fuel, in the long run, the conventional anode of SOFC (Ni–YSZ) undergoes carburization in presence of HC fuels, which is catastrophic for the performance of SOFC. Very recently, a large number of reviews have been published on hydrocarbon-based SOFCs [11–17]. Dewa et al. [12] have reviewed the reforming catalysts for metal-supported SOFC. Wei et al. [13] have provided an account of perovskite materials for reforming $CH_4$ in SOFCs. In a very recent review, Liu et al. [14] have summarized the fundamentals and challenges in using hydrocarbons directly in the upcoming proton-conducting SOFCs. Shabri et al. [15] have presented the strategies such as alloying and combining the ceramic component with mixing oxygen carrier that contains perovskite for using a cermet material as an anode to overcome the carbon deposition. Zhang et al. [16] have presented the progress of the catalyst layer materials for hydrocarbon-fueled SOFC and issues related to the use of those layers. Su et al. [17] have focused on the challenges and strategies associated with electrolytes, anodes, and cathodes for low-temperature SOFC. Shi et al. [18] have provided an account of anodic reactions in HC-based fuels and also discussed the properties and models of novel oxide anodes such as in situ exsolved metal catalysts.

The current review focuses on the issue of carburization and strategies adopted to control carburization. The underlining goal of this review is to discuss various alternate HC-compatible SOFC anode catalysts and their synthesis techniques. The issue of sulfur poisoning and strategies to control it are also presented.

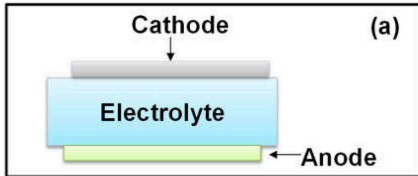
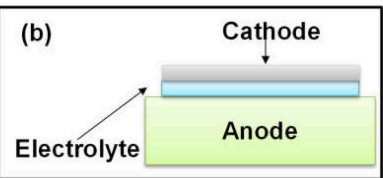

**Figure 1.** Schematic of most popular SOFC planar configurations: (**a**) ESC and (**b**) ASC.

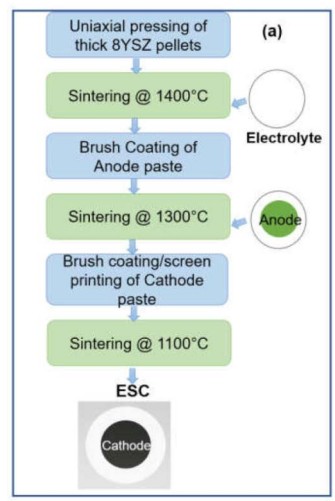
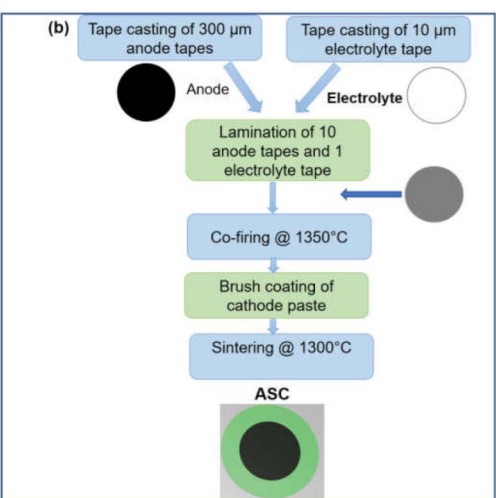

**Figure 2.** Flowchart depicting the steps involved in the fabrication of (**a**) ESC and (**b**) ASC.

## 2. Utilization of Hydrocarbon Fuels in SOFC

In the past two decades, direct internal reforming (DIR) in SOFC has been a topic of acute research. To circumvent the use of external reformers, efforts have been made to employ HCs directly in SOFCs. Since they are operated at high temperatures, the DIR of HC fuels can happen in the anode itself. While employing HCs as the fuel, it is generally assumed that SOFCs are operating on syngas (CO + $H_2$) formed during reforming. Because of the high operating temperatures of SOFCs, HCs can be supplied to the anode right away without reforming externally. Therefore, the direct use of HCs could tremendously enhance fuel cell efficiency by evading the losses involved with external reformers. Higher molecular weight HCs undergo various elementary reactions at SOFC operating conditions. The elementary reactions are generally classified as (i) steam reforming, (ii) auto thermal reforming, (iii) partial oxidation (POX), and (iv) direct methane oxidation [19,20]. The reactions involved, along with their advantages and limitations, are summarized in Table 1.

Among the above reactions, DM-SOFCs possess maximum catalytic activity for the complete oxidation of methane. Further, complete electricity generation is possible without syngas cogeneration by using an anode catalyst that fosters $CO_2$ and $H_2O$ production. It is mostly agreed that the carbonaceous adsorbates are easily oxidized by the chemisorbed oxygen species present on the metal surface. The surface oxygen for oxidation is made available from oxidants ($H_2O$, $CO_2$, and $O_2$) or the lattice oxygen provided by oxide supports. Therefore, it is possible to control the carburization by controlling the reaction by feed composition. However, the present review is dedicated to understanding the basic ability of anode catalysts in controlling carburization.

### 2.1. Carburization

HCs cannot be utilized directly in the state-of-the-art SOFCs as Ni-based anodes are not stable when HCs are employed as fuel due to the formation of carbon fibers, and details are presented in the next section.

**Table 1.** Different reactions of HCs occurring at the operating conditions of SOFC along with their advantages and limitations.

| Types of Reactions | Reactions Involved | Temp. Range (°C) | Advantages | Limitations |
|---|---|---|---|---|
| Steam Reforming | $CH_4 + H_2O \Leftrightarrow CO + 3H_2$ (syngas) (endothermic) $CO + H_2O \rightarrow CO_2 + H_2$ (exothermic) | 700 | Suitable for external reformers | Reforming fuel inside SOFC anode (internal steam reforming) will affect the performance; more steam results in low OCV in SOFC due to fuel dilution |
| Autothermal Reforming (ATR) | $2CH_4 + O_2 + CO_2 \rightarrow 3H_2 + 3CO + H_2O$ (exothermic) $4CH_4 + O_2 + 2H_2O \rightarrow 10H_2 + 4CO$ | 950–1060 | Reactor is compact in design | Reaction pressure is in the range 30–50 bar |
| Partial Oxidation (POX) | Thermal partial oxidation (TPOX) $C_nH_m + \frac{2n+m}{4} O_2 \rightarrow n\,CO + \frac{m}{2} H_2O$ (exothermic) Catalytic partial oxidation (CPOX) $C_nH_m + \left(\frac{m}{2} + n\right) O_2 \rightarrow n\,CO + \frac{m}{2} H_2O$ (with catalyst) | 1200 | CPOX reaction utilizes a catalyst to reduce the reaction temperature to around 800–900 °C | CPOX is suitable only for low sulfur (<50 ppm) fuel as catalysts are much more prone to sulfur poisoning. CPOX has seldom been used in SOFC anodes, as the pO$_2$ in the chamber must be sustained <$10^{-18}$ atm to trigger SOFC operation |
| Direct Methane (DM) Oxidation | $CH_4 + 2O_2 \rightarrow CO_2 + 2H_2O$ | >750 | DM-SOFCs possess maximum catalytic activity for complete oxidation of methane | Oxidation of Ni |

### 2.1.1. Carburization Mechanism

The mechanism of carbon fiber formation involves the following steps: (i) carbon from the HC is deposited on the Ni surface, (ii) it undergoes dissolution into the bulk of the Ni, and (iii) it is precipitated as a fiber. The carbon fiber formation leads to Ni loss due to a process called "metal dusting", wherein Ni atoms are physically picked from the surface due to their entanglement with the growing carbon fibers. The metal dusting starts with the transfer of carbon from the carbon supersaturated environment (Carbon activity ($A_c$) > 1) to the metal surface. Subsequently, graphitic carbon deposits on the metal surface and forms graphitic planes (often perpendicular to the metal surface), resulting in the formation of a kind of channel between planes for transferring metal ions. Finally, the detached metal particles catalyze the filamentous carbon [21]. Due to the growth of carbon fibers, mechanical stresses are generated in the SOFC, resulting in its fracture [22–24].

### 2.1.2. Carburization Kinetics

Basically, three types of reactions are considered as the source of carbon deposition in the operating condition of SOFC while using methane and CO as fuels. They are

(i)  Methane cracking

$$CH_4 \rightarrow C + 2H_2 \quad \Delta H_{298K} = +19 \text{ kJ mol}^{-1} \tag{1}$$

(ii)  Reduction of carbon monoxide

$$CO + H_2 \rightarrow C + H_2O \quad \Delta H_{298K} = -131 \text{ kJ mol}^{-1} \tag{2}$$

(iii)  Boudouard reaction

$$2CO \rightarrow C + CO_2 \quad \Delta H_{298K} = -172 \text{ kJ mol}^{-1} \tag{3}$$

Figure 3a shows that the Boudouard reaction is highly active below 800 °C and fades away above 800 °C, whereas, from Figure 3b, it is evident that the methane cracking reaction is favored above 800 °C [25]. Further, the equilibrium concentration of typical feed gas of SOFC over the temperature range of 0–1000 °C indicates minimal carbon deposits

at 800 °C (Figure 4) [26]. However, it is difficult to decide upon the vulnerable conditions based on just the temperature and gas composition.

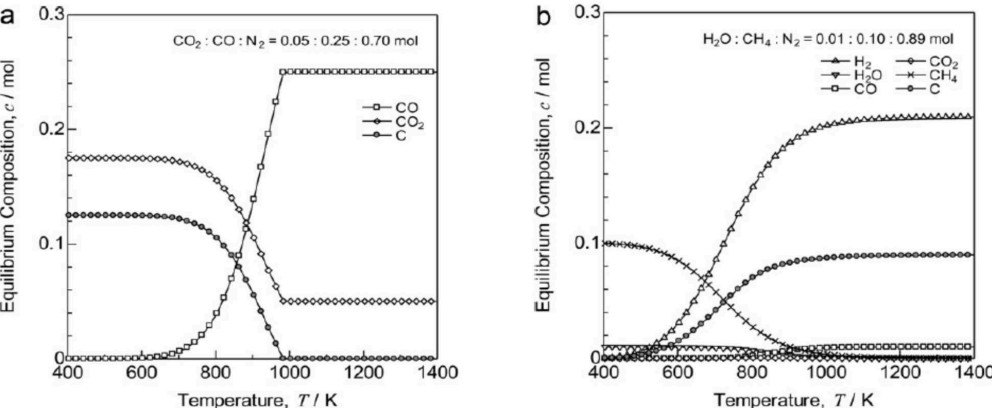

**Figure 3.** The equilibrium concentration of product for a typical SOFC feed composition (**a**) $CO_2$:CO:$N_2$ = 0.05:0.25:0.70 and (**b**) $H_2O$:$CH_4$:$N_2$ = 0.01:0.10:0.89 mol (adapted from [25]).

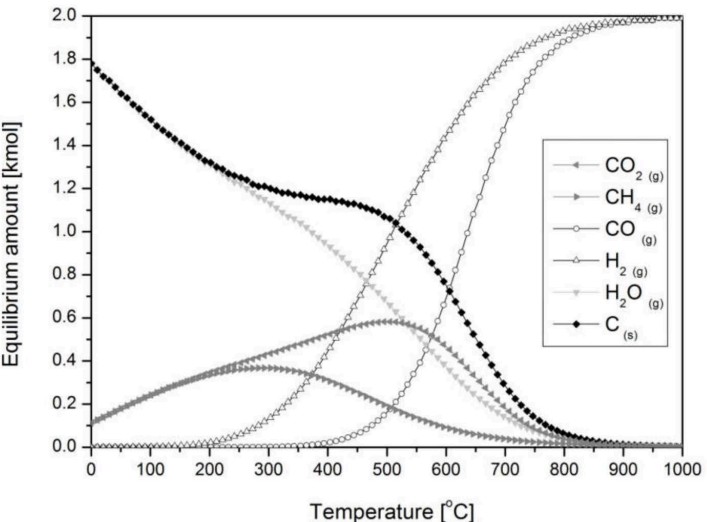

**Figure 4.** Equilibrium amounts for gaseous components and coke in the dry reforming of methane process ($CH_4$/$CO_2$/Ar = 1/1/8, $p$ = 1 bar) (adapted from [26]).

The carbon activity decreases with increasing temperature while the diffusivity of carbon in metal and carbon saturation concentration increases with temperature. Thus, in addition to temperature and gas composition, material property plays a key role in deciding the vulnerable condition. Dean et al. [27] have derived the following parameter to quantify the vulnerability based on material property and carbon activity:

$$N_c = D_c X_c (1 - A_c) \qquad (4)$$

where

$N_c$—parameter to quantify carburization vulnerability
$D_c$—diffusivity of carbon in metal at a given temperature
$X_c$—maximum solubility of carbon in the metal
$A_c$—carbon activity.

The diffusion constant is directly proportional to the temperature. As the diffusion constant decreases to a greater extent than the increasing carbon activity with decreasing temperature, there is no impact of higher carbon activity felt at low temperatures. Diffusion is driven by the concentration gradient of carbon at the surface to the bulk of the metal. In

order to convert carbon activity to a concentration of carbon, it is necessary to multiply it by the saturation concentration of carbon in metal. Thus, the diffusion coefficient, the carbon concentration and the activity coefficient are good indicators of the driving force to cause carbon to diffuse into the alloy.

### 2.1.3. Strategies to Control Carburization

So far, numerous approaches have been attempted to control or curtail the carburization issue. Most of those approaches can be classified as shown in Figure 5.

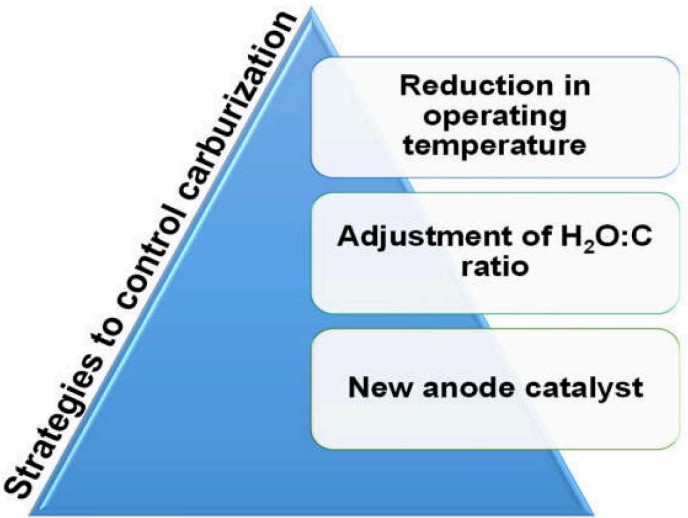

**Figure 5.** Strategies to control carburization.

The first strategy to control carburization is reducing the operating temperature of SOFC. It is evident from Equation 4 that the diffusivity and carbon saturation concentration would be lower at the lower operating temperature of the SOFC (below 700 °C). However, the oxygen ion conduction also reduces with temperature. Further, the carbon activity increases with decreasing temperature. Hence, the SOFC has to be operated at a relatively higher current density to check the carbon deposition, which in turn would lead to high concentration polarization loss. Hence, this strategy may not be effective.

Secondly, the problem of carbon fiber formation on Ni can be circumvented by enriching the fuel with adequate amounts of steam to enable the removal of carbon at a faster rate compared to its deposition rate. The presence of steam would enable the methane steam reforming reaction in the anode and reduce the carburization. Usually, thermodynamic analysis is employed to anticipate the $H_2O:C$ ratios essential to circumvent carbon fiber formation. However, fibers are formed at much higher $H_2O:C$ ratios than the predicted values. This clearly shows that the stability of Ni anodes is dependent on the comparative rates of carbon deposition and its diminution.

The rate of carbon deposition with $CH_4$ is comparatively low and, hence, $CH_4$ can be internally reformed in SOFC by maintaining the $H_2O:C$ ratio as unity. Since carbon deposits faster on Ni in presence of higher HCs, higher $H_2O:C$ ratios have to be maintained to prevent carbon fibers formation. In this scenario, a higher amount of steam will result in fuel dilution and thereby reduce the SOFC efficiency. Further, steam enhances the formation of $Ni(OH)_2$, which in turn destabilizes the anode. Above all, the thermal stress induced by the endothermic steam reforming reaction can damage the cell. This approach has not been widely used due to the destructive aftermath of fiber formation such as cell fracture and removal of Ni [28,29].

Another approach for direct utilization of HCs as fuel in SOFCs is to choose alternate anode materials that do not activate the formation of carbon fibers similar to Ni. The solubility of carbon in Cu, Ag, and Au is lower than Ni, Fe, Co, and Ru and does not result in fiber formation [30,31]. However, in general, polyaromatic species generated by

gas-phase pyrolysis can deposit on the surface of any material; fortuitously, these deposits are not as damaging as that of fiber formation. Since the condensed carbon is confined at the electrode surface, it may not disintegrate the anode as the fiber does. These deposits can be avoided easily by providing a catalytic coating such as ceria on the surface. The ceria coating in the presence of an adequate quantity of steam catalyzes the oxidation of the deposits [32]. Furthermore, the disadvantage with most of the prospective anode materials is that their effectiveness has not equipoised the best Ni anodes. The main issue with metal-based Cu anodes is that they are sintered, resulting in relatively poor thermal stability [33,34]. The important disadvantages of ceramic anodes are their low conductivity and poor catalytic activity [35]. Thus, it becomes essential to develop suitable anode composition to control the carburization without compromising the performance of SOFC. The alternate anode has to ensure better catalytic activity towards various anode reactions such as hydrogen oxidation, steam reformation, and POX and ATR reactions. However, as a basic criterion, the anodes are being evaluated for the catalytic activity of hydrogen oxidation reaction by most researchers [36,37]. In the subsequent section, the kinetics of the electrochemical oxidation of fuel at the anode is discussed.

## 3. Kinetics of Electrochemical Oxidation of Fuel at Anode

The overall reaction occurring at the anode for a simple hydrogen oxidation reaction is as follows:

$$O_o(\text{electrolyte}) + H_2 \text{ (fuel gas)} \rightarrow H_2O \text{ (g)} + 2e^\bullet + V_o^{\bullet\bullet} \qquad (5)$$

Though the hydrogen oxidation reaction appears to be simple, various intricate elementary reaction pathways were proposed by different researchers, and it is needless to mention the complexity involved in the reactions associated with the utilization of HCs. The heterogeneous catalytic reaction that occurs at the SOFC anode is a Langmuir–Hinshelwood type reaction in which the adsorbed reactants undergo surface reaction; subsequently, the product is desorbed from the surface. Ihara et al. [38] experimentally verified the Langmuir reaction model and linked the dependence of electrical properties of the anode with the chemical reactions.

High catalytic activity for these surface reactions is achieved by having optimum strength of chemisorption between reactants and metal surfaces. Higher adsorption strength would poison the surface, whereas the lower strength would result in starvation for reactants. Therefore, the volcano type curve is the best descriptor for catalytic activity as a function of adsorption strength. For instance, in the case of $H_2$ oxidation, the adsorption energy of reactants such as $H_2$, $O_2$, or $OH^-$ would dictate the reaction rate.

### 3.1. Electrochemical Oxidation of $H_2$

Based on the kinetics involved, the elementary reaction mechanisms are categorized into six different types such as (i) oxygen spillover, (ii) hydroxyl spillover, (iii) hydrogen spillover, (iv) interstitial hydrogen transfer, (v) reactive electrolyte, and (vi) reverse cathode [39,40]. All these electrochemical reaction mechanisms mainly concur with the adsorption and desorption behavior of $H_2$ and $O_2$ and the formation of $OH^-$. The major dissimilarity is the site where the chemical and the electrochemical reactions take place, as well as the charge transfer step.

### 3.1.1. Electrocatalytic Activity of Metals

One of the main criteria that play a critical role in deciding the type of elementary kinetics is the material property, such as adsorption energy and diffusivity of metallic anode for a given reactant. Rossmeisl et al. [36] have correlated the density function theory (DFT)-derived adsorption energy of chemisorbed species on various metal surfaces to the experimentally observed conductivity. The results suggested that the $O_2$ spillover

mechanism played a dominant role in anode catalysis. The elementary reaction steps are as follows [36]:

$$O^{2-} \Leftrightarrow O^* + 2e^-$$
$$\Delta G_1 = G_O = \Delta E_O + 0.02 \text{ eV}$$

(6)

$$O^* + H_2 \Leftrightarrow OH^* + 1/2\,H_2$$
$$\Delta G_2 = G_{OH} - G_O = \Delta E_{OH} + 0.85 \text{ eV} - \Delta E_O - 0.02 \text{ eV}$$

(7)

$$OH^* + 1/2\,H_2 \Leftrightarrow H_2O$$
$$\Delta G_3 = -G_{OH} = -\Delta E_{OH} - 0.85 \text{ eV}$$

(8)

It was reported that metals such as Ni, Co, Rh, Ru, and Ir can exhibit superior catalytic activity for $H_2$ oxidation reaction (Figure 6).

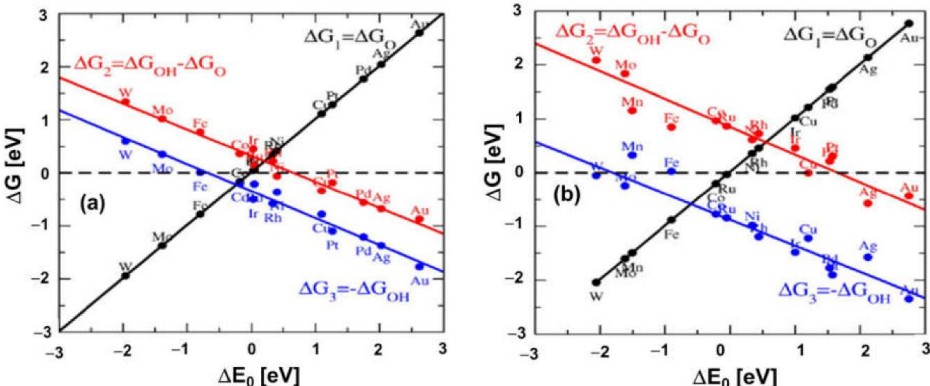

**Figure 6.** Gibbs free energy for the individual elementary reactions of the anode for different metal catalysts. (**a**) Values for surface terraces and (**b**) for surface steps. The black, red, and blue lines correspond to the change in free energy for the Equations (6)–(8), respectively (adapted from [36]).

Literature also entails other types of spillover mechanisms such as $H_2$ and hydroxyl. DFT analysis on the hydrogen spillover mechanism was reported by Ingram et al. [37]. The trend was very similar to the $O_2$ spillover mechanism [37]. Lykhnytskyi et al. [41] also discussed the interrelation within the exchange current of hydrogen ion reduction and bond energy [41]. In either case (oxygen or hydrogen spillover mechanism) metals such as Ni, Co, Rh, Ru, and Ir showed higher catalytic activity. Additionally, they are proven to be active catalysts for CO oxidation and methane steam reforming reactions. It is evident from the literature [36,41–45] that it is difficult for any metal to match the properties of Ni in the stringent SOFC operating conditions. Cu is considered a potential candidate to alloy with Ni due to its inability to catalyze C–C bond formation. Similarly, Co is considered due to its lower carbon diffusivity (1/4th of Ni). It is evident from Equation (4) that the carbon diffusivity plays a critical role in carburization. Further, rare earth oxides are also incorporated into the system to catalyze the reforming reaction. Another important factor that influences kinetics is the ionic conductivity of anode cermet [42]. More details on this are presented in the next section.

### 3.1.2. Oxygen Ionic Conductivity of Anode Composite

The intrinsic charge transfer resistance for a given electrocatalyst/electrolyte pair is given by Equation (9). In presence of ionic conductivity, the reaction zone spreads out from the electrode/electrolyte interface to the inner part of the electrode. Therefore, it is essential to have a good ionic conducting phase to reduce the charge transfer resistance. The effective charge transfer resistance for a reasonably good ionic conductor such as YSZ can be given by the simplified Tanner equation (Equation (9)) [42].

$$R_{ct}^{eff} \approx \sqrt{\frac{BR_{ct}}{\sigma_i(1 - V_v)}}$$

(9)

where B—grain size of electrolyte (μm), $V_v$—fractional porosity, $\sigma_i$—ionic conductivity, $R_{ct}^{eff}$—effective charge transfer resistance, and $R_{ct}$—charge transfer resistance.

$$R_{ct} = \frac{RT}{nFi_o} \tag{10}$$

where R—gas constant; n—number of electrons, F—Faraday constant, and $i_o$—exchange current density.

Figure 7 shows that, in the case of the anode with higher ionic conductivity, charge transfer resistance decreases with anode thickness, whereas, at a lower ionic conductivity, charge transfer resistance increases with anode thickness. This is due to the fact that reaction kinetics is restricted to the electrolyte anode interface in the latter case. Therefore, the ionic conducting phase in the anode cermet enhances the reaction kinetics.

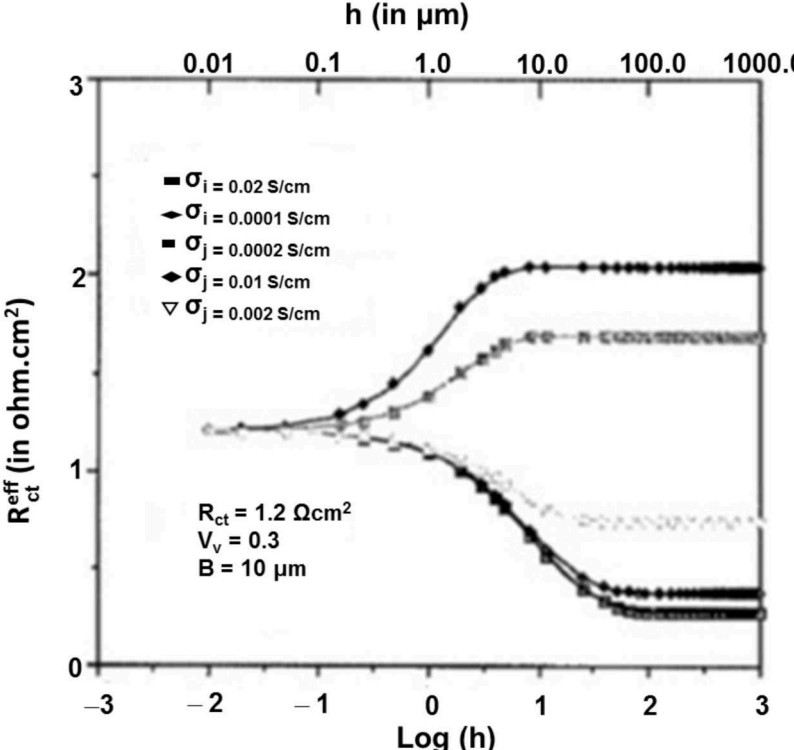

**Figure 7.** Correlation between effective charge transfer resistance and electrode thickness of different ionic conductivity (adapted from ref. [42]).

There were also attempts to enhance the performance of anode reaction kinetics by employing a mixed ionic electronic conductor (MIEC). However, the thermal, chemical, and mechanical stability of Ni–YSZ made it more appealing for high-temperature SOFC. In fact, the carburization-related issues are critical for Ni–YSZ in the HC fuel environment, which still needs to be addressed.

### 3.2. Electrochemical Oxidation of Carbon Monoxide

Carbon monoxide (CO) is not considered a favorable fuel for SOFC because of its considerably higher anodic overvoltage [46–49] and more sluggish oxidation kinetics than $H_2$ [50]. Three different charge transfer mechanisms based on the oxygen spillover mechanism were predicted for the CO oxidation process and compared with experimental results by Yurkiv et al. [51]. The results obtained by numerical simulations of the above reaction mechanism were compared with experimental data obtained by Lauvstad et al. [52,53].

Amidst concerns of carbon deposition, Jiang and Virkar [46] demonstrated cells with an output of 0.7 $Wcm^{-2}$ at 800 °C when operated with CO as a fuel. The calculated

standard Gibbs energy for CO and $H_2$ oxidation at 850 °C operating temperature was $-185$ kJ $mol^{-1}$ of $CO_2$ vs. $-186$ kJ $mol^{-1}$ of $H_2O$, respectively. Additionally, the enthalpy change for the CO oxidation reaction was lightly more exothermic ($-282$ kJ $mol^{-1}$ for $CO_2$) than for the oxidation of $H_2$ ($-249$ kJ $mol^{-1}$ for $H_2O$). The reaction contributing to carbon deposition in CO fuel is the Boudouard reaction [25]. Nevertheless, it is suppressed above 700 °C. Hence, carburization is expected to be low even for an Ni–YSZ anode at a high operating temperature. Homel et al. [54] operated a 50-cell stack for about 375 h without any signs of degradation at 850 °C with CO.

### 3.3. Electrochemical Oxidation of Hydrocarbon

In a typical SOFC anode, HC fuels can take part in a wide range of reactions such as direct electrochemical oxidation, various reforming processes, and surface-catalyzed carbon deposition. In the literature, the reforming kinetics have been well-established for industrial-scale packed/fluidized bed reactors. Varying reaction mechanisms and related kinetic models have been proposed based on steam reforming, dry reforming with $CO_2$, total, and POX [55–58]. In addition, mechanistic models based on principles of the elementary steps and their energetics were proposed [58–60]. In the case of SOFC, the microstructure of the anode is instrumental in reforming kinetics. Further, the catalyst-support chemistry may vary with YSZ or ceria-based cermets. Evidence shows that the YSZ possesses catalytic activity for POX [61], and ceria is good at electrochemically oxidizing carbon deposits [62].

The reaction mechanism describing the heterogeneous kinetics comprising of 42 irreversible reactions involving six gas-phase and 12 surface-adsorbed species within a Ni–YSZ anode is well-documented in the literature [63]. The reaction rates of these elementary steps are portrayed in the Arrhenius form or as a sticking coefficient. Since the reaction mechanism is centered on elementary molecular processes, it illustrates all the ubiquitous processes in a SOFC anode such as (i) steam reforming of $CH_4$ to CO and $H_2$; (ii) water–gas shift processes; and (iii) surface carbon coverage.

Laosiripojana et al. [64] investigated the effect of SOFC operating temperature and the inlet steam content on the quantity of carbon formation due to the steam reforming of HCs at 900–1000 °C for varying inlet fuel/steam molar ratios. The influence of the inlet steam content and temperature on all product distribution and quantity of carbon formation using fuels such as $CH_4$, $CH_3OH$, and $C_2H_5OH$ is well-documented in the literature [64]. With an increase in the temperature and inlet steam concentration during steam reforming of $CH_4$ and $CH_3OH$, the amount of carbon formation decreased drastically. This may be attributed to their higher reforming reactivity at high temperatures and inlet steam concentrations. Further, the $H_2$ and $CO_2$ fraction raised with increasing inlet steam concentration, while the CO fraction decreased. However, high steam concentrations are known to increase the sintering of the nickel in addition to reducing the open circuit voltage (OCV) and thermodynamic efficiency of the cell [65]. The DIR of HC in SOFC anode offers several advantages compared with external reforming [65]: (i) no separate steam reforming unit is required, and, as a result, the system cost is reduced; (ii) DIR reduces the steam requirement; (iii) evenly distributed load of $H_2$ in a DIR SOFC leads to a more homogeneous temperature distribution; and (iv) higher methane conversion.

A large number of HCs such as LPG, propane, naphtha, etc., and methanol can be used as fuels. However, methane ($CH_4$), also termed natural gas, is the preferred fuel. It has been shown that the HCs' steam reformation over nickel occurs through surface carbon species [66]. Chemisorption of HCs on metals includes the direct scission of a C—H bond. Excluding methane, almost all HCs assume a two-site mechanism wherein the adsorbed molecule is not required as the precursor. For methane conversion over nickel, an adsorbed CH species is changed to an adsorbed carbon atom via sequential dehydrogenation:

$$CH_4 = CH_3^* \rightarrow CH_2^* \rightarrow CH^* = C \tag{11}$$

In methane steam reforming, it has been suggested that methane adsorption is the rate-determining step that is in accordance with the mainly assumed first-order dependency of the rate in $CH_4$.

Some researchers disagreed with single-step electrochemical oxidation of HC and provided detailed insight into the electrochemical oxidation of HC in SOFC anode [62]. Even the simplest HC, methane, is expected to have the following elementary steps during direct electrochemical oxidation.

$$CH_4 + O^{2-} \rightleftharpoons CH_3OH + 2e^- \tag{12}$$

$$CH_3OH + 2O^{2-} \rightleftharpoons HCOOH + H_2O + 4e^- \tag{13}$$

$$HCOOH + O^{2-} \rightleftharpoons CO_2 + H_2O + 2e^- \tag{14}$$

Similarly, HC cracking accompanied by electrochemical oxidation should involve following elementary steps:

$$CxHy \rightleftharpoons xC + y/2H_2 \tag{15}$$

$$C + 2O^{2-} \rightleftharpoons CO_2 + 4e^- \tag{16}$$

$$H_2 + O^{2-} \rightleftharpoons H_2O + 2e^- \tag{17}$$

Mogensen and coworkers [62] stressed the need to differentiate among the reaction pathways as the requirements on the anode material vary remarkably. Therefore, the needs of the properties of the anode catalyst are basically dissimilar. In principle, both pathways are possible in the anode. The operating temperature would be the deciding factor of the dominant pathway, and it would be desirable to operate below cracking temperature. Hence, it is desirable to have a catalyst with good activity for direct electrochemical oxidation with some resistance to carburization. Accordingly, the subsequent section is devoted to the identification of carbon tolerant catalyst for electrochemical oxidation of dry $CH_4$.

The detailed scrutiny of anode kinetics suggests that there is a dire need for improving the SOFC anode design, and, to achieve the high catalytic activity, research has to be focused on maximizing the reactive area and elementary reaction kinetics at the anode/electrolyte interface. Several researchers have focused their research on fabricating HC-compatible anodes, and details are presented below.

## 4. Hydrocarbon (HC) Compatible Anodes

New anode developments are focused on carbon-tolerant ceramic oxide and bimetallic anode systems fulfilling the critical requirements of SOFC. The anode of SOFC plays various roles such as (i) hosting triple-phase boundaries (TPB) to support electrochemical reaction, (ii) providing ionic and electronic conducting paths, and (iii) providing channels for gaseous reactants. Further, anodes also provide mechanical support in the case of ASC. Figure 8 shows that there is a rising trend in R&D on HC-based SOFC in the recent decade as per the Scopus analysis carried out. More specifically, there are substantial efforts to develop new anode compositions suitable for HC fuels.

The leading authors who have contributed significantly in the area of HC-based SOFCs are Prof. R.J. Gorte, Prof. J.M. Vohs, Prof. JTS Irvine, Prof. Luo, etc. The top countries pursuing research in this area are the USA, China, Japan, South Korea, UK, Canada, etc. The leading institutes involved in HC-based SOFC research are Colorado School of Mines, University of Pennsylvania, Kyushu University, University of South Carolina, Chinese Academy of Sciences, etc.

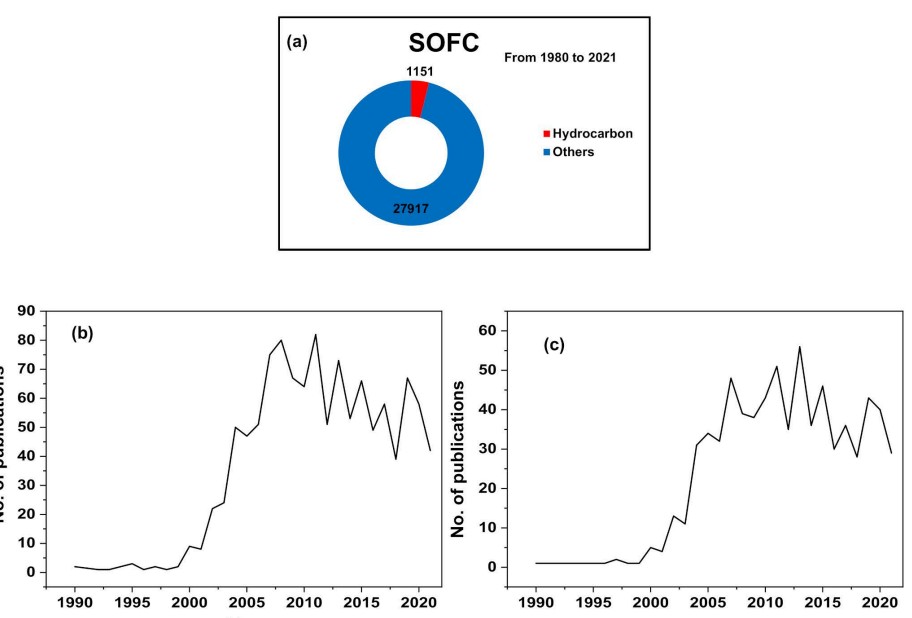

**Figure 8.** Scopus database analysis using keywords (**a**) SOFC, (**b**) SOFC+ HC, and (**c**) SOFC +HC+anode (data were acquired on 13 October 2021).

A large variety of materials are studied as anodes for HC-based SOFCs. These anode materials are discussed under the headings shown in the flowchart (Figure 9).

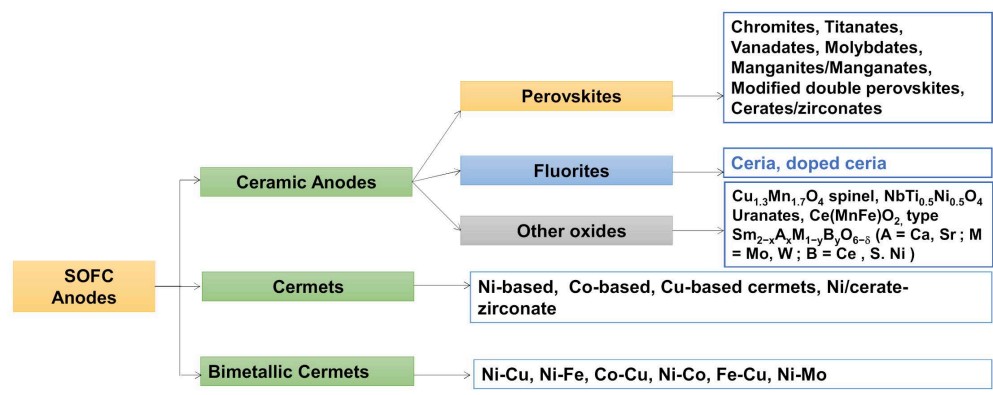

**Figure 9.** Flowchart showing the classification of HC-compatible SOFC anodes.

## 4.1. Ceramic Anodes

Ceramic oxide anodes such as lanthanum strontium chromium manganates, titanates, and vanadates are known for their carbon-retarding ability, high temperature, and redox stability. Unlike metals, these are not the aggressive catalysts for C–H scission instead assist in the direct electrochemical oxidation of HC fuels [66–68]. Accordingly, it limits the temperature gradient across the electrode. The main disadvantage of the oxide anodes is their low catalytic activity. Often, they are doped or used in cermet composite to achieve the required performance. The different types of ceramic anodes reported in the literature are discussed in more detail below.

### 4.1.1. Perovskite Structure

Among the ceramic oxide anodes, perovskites have found a prominent place. Because of their capability to accommodate a wide range of A site and B site cations, properties can be tailored to match the anode requirements. The recent developments on perovskite oxide-based electrocatalysts and their applications in electrochemical devices and oxygen-related

electrocatalysts have been well-documented [69,70]. Perovskite oxides such as lanthanum strontium chromites (LSC), lanthanum strontium titanates (LST), lanthanum strontium vanadates (LSV), and their doped derivatives are well-studied in the literature, and more details are presented below.

Chromites

Lanthanum chromite ($LaCrO_3$ and LC) is inert towards methane oxidation reaction. However, the addition of Ca and Sr to the A site improves the catalytic activity. Similarly, the addition of Mn and Ni to the B site drastically enhances the catalytic activity [71].Barison et al. [72] explored the catalytic property of $Au/La_{1-x}Sr_xMnO_3$ ($x \approx 0.4$) and $Au/La_{0.85}Sr_{0.15}CrO_3$. The catalytic ability for propane reforming through different modes of reaction such as steam reforming, POX, and ATR were studied, and the propane conversion was up to 100% in the POX condition, despite weak phase stability. On the contrary, the LSC had displayed weak hydrogen selectivity. Additionally, it was established that the reaction was not influenced due to the presence of gold [72]. The electrical conductivity of $(La_{0.75}Sr_{0.25})(Cr_{0.5}Mn_{0.5-x}Ni_x)O_{3-\delta}$ and $(La_{0.75}Sr_{0.25})(Cr_{0.5-x}Ni_xMn_{0.5})O_{3-\delta}$ anodes slightly increased with the Ni addition up to x = 0.06. However, further addition of Ni content reduced the conductivity [73]. Though these compositions are found to be good for electrolyte-supported cells, the electrical conductivity was low for anode-supported cells. $La_{0.75}Sr_{0.25}Cr_{0.5}Co_{0.5}O_{3-\delta}$ exhibited an electrical conductivity of 133.8 $Scm^{-1}$ but the high TEC of $19.5 \times 10^{-6}$ $^{\circ}C^{-1}$ ruled out its usage in SOFC [74]. By doping LSC with Ni, the power density of SOFC was increased up to 180 $mWcm^{-2}$ [75]. However, the power density reduced after 190 h. On the other hand, Ru-doped LSC exhibited a power density of 300 $mWcm^{-2}$. Even though Ru-doped LSC exhibited better performance, it would be expensive to induct Ru in anodes [75]. The performance of $La_{0.8}Sr_{0.2}Cr_{1-x}Ru_xO_{3-\delta}$ (LSCrRu) anode increased with time for smaller x. Nevertheless, the performance dropped with time for higher $\times$ values [76]. Ru-doped LSC was not promising as it exhibited lower electrical conductivity and Sr segregation [77–79]. The A-site-deficient LSCNi improved the exsolution of Ni nanoparticles on the surface of the anode catalyst and resulted in the enhancement of anodic performance and catalytic activity. Sun et al. [80] reported favorable electrochemical performance (460 $mWcm^{-2}$ at 800 $^{\circ}C$) and redox stability for A-site-deficient $La_{0.6}Sr_{0.3}Cr_{0.85}Ni_{0.15}O_{3-\delta}$ in 5000 ppm $H_2S-H_2$.

Titanates

Even though titanates are known to possess good electronic conductivity and lower ionic conductivity, they are sluggish in anode catalysis. Sinha et al. [81] explored the use of reaction-sintered titanium oxycarbide ($TiO_xC_{1-x}$ with x = 0.2–0.8) under vacuum at 1500 $^{\circ}C$ for 5 h as a viable rare-earth free anode material for IT SOFCs. The $TiO_{0.2}C_{0.8}$ anode was stable, and reaction with GDC was observed at 900 $^{\circ}C$ in Ar + 5%$H_2$ atmosphere. The GDC electrolyte-supported SOFC consisting of a $TiO_{0.2}C_{0.8}$ anode exhibited a maximum power density of 130 $mWcm^{-2}$ at 700 $^{\circ}C$. The lower ionic conductivity was addressed to some extent by doping. Miller et al. [82] studied B-site doped lanthanum strontium titanate (LST). Some dopants such as Fe and Mg reduced the valency of Ti and resulted in poor performance [82]. Rare earth doping increased the ionic conductivity of titanates. Scandium doping also improved the ionic conductivity of LST. However, $La_{0.3}Sr_{0.7}Sc_xTi_{1-x}O_{3-\delta}$ is found to have ionic conductivity at the cost of total electrical conductivity with the addition of Sc. Despite all these, the ionic conductivity was four times lower than that of YSZ [83]. In Co-doped LST, as the Co content increased, the electrical conductivity reduced. Further, the electrical conductivity also drops with the temperature. However, the ionic conductivity was 0.006 $Scm^{-1}$ at 700 $^{\circ}C$ and increased with temperature [84]. Therefore, it was proposed in the context of intermediate temperature SOFC (IT-SOFC). Similarly, cobalt-doped $Y_{0.08}Sr_{0.92}TiO_{3-\delta}$ exhibited improved ionic conductivity but the electronic conductivity was reduced. The addition of cobalt increased the operating temperature due to the improved structural stability [85].

Périllat-Merceroz et al. [86] reported layered $La_xSr_{1-x}TiO_{3+\delta}$ (LST) (LST2D), which performed better than the classical 3D LST perovskite (LST3D). LST2D showed 10 times more $H_2$ generation compared to that of LST3D. However, the electrical conductivity was about two orders of magnitude lower than LST3D [86]. LST was doped with ceria and noble metals to enhance the anode catalysis. Lanthanum strontium titanium manganite ($La_{0.4}Sr_{0.6}Ti_{0.8}Mn_{0.2}O_{3-\delta}$, LSTM), showed relatively lower area-specific resistance and stable performance in hydrogen and methane at high temperatures [87,88]. The power density of LSTM increased by seven-fold with the addition of 10% $CeO_2$ and 1% Pd, 10% $CeO_2$ increased the power density by five-fold, and 1% Pd enhanced the power density by two-fold [89]. Nonstoichiometric strontium yttrium titanates (SYT, 2–6% A site defective) exhibited better conductivity (100 $Scm^{-1}$), whereas B-site-defective SYT exhibited poor conductivity in the range of 0.01 $Scm^{-1}$ [90]. Though SYT displayed good performance, its conductivity is sensitive to the oxygen partial pressure and decreases with an increase in $P_{H2O}$. The increasing ohmic resistance in the presence of steam restricts SYT performance in internal reforming conditions [91,92]. The addition of metals such as Ni, Cu, Ru, etc., has improved the anode catalysis [93–95]. Miao et al. [96] reported an increase in ionic conductivity of $Sr_{0.88}Y_{0.08}TiO_3$ due to doping of Yb in the A-site and B-site. Among them, $Sr_{0.88}Y_{0.06}Yb_{0.02}TiO_3$ displayed the highest ionic conductivity and power density (87 $mWcm^{-2}$) under $CH_4$ at 800 °C. Cao et al. [97] explored $La_{0.7}Sr_{0.3}Fe_{0.7}Ti_{0.3}O_3$ (LSFT) as an SOFC anode material with an ESC configuration of LSFT|SDC|YSZ|LSM/YSZ using $CH_4$ and $H_2/H_2S$ as fuels. The SOFC displayed a consistent maximum power density of 121 $mWcm^{-2}$ at 850 °C for 24 h with humidified methane fuel and was free from carburization problems. Błaszczak et al. [98] synthesized La-, Ce-, and Ni-doped LST ($La_{0.27}Sr_{0.54}Ce_{0.09}Ni_{0.1}Ti_{0.9}O_{3-s}$ (LSCNT)) using the Pechini method followed by reduction at 900 °C, resulting in exsolved Ni ions at the surface of the LSCNT in the form of spherical nanoparticles. LSCNT was applied on the anodic side of the SOFC NiO–YSZ/YSZ/LSM-YSZ cell, which resulted in increased stability and catalytic activity in the presence of the synthetic biogas stream. Another anode with the composition $La_{0.875}Sr_{0.125}Ti_{0.5}Ni_{0.5}O_3$ (25LSTN50) was used for the development of symmetrical cells [99]. Though it exhibited a similar electrochemical performance to that of materials reported in the literature used in symmetrical cells, the polarization resistance was one order of magnitude greater than the reported values [99]. Arrivé et al. [100] studied the stability of $La_{2x}Sr_{1-2x}Ti_{1-x}Ni_xO_{3-\delta}$ (LSTN) and $La_{7x/4}Sr_{1-7x}/4Ti_{1-x}Ni_xO_{3-\delta}$ (25LSTN) materials in SOFC conditions with greater emphasis on the Ni exsolution process for anode application. A molecular dynamics (MD) model was developed and applied for optimizing the sintering of La-doped $SrTiO_3$ (LST) and gadolinium-doped ceria (GDC) and surface diffusion was found to be the main sintering mechanism [101].

Another promising titanate-based electrode material is yttria-doped strontium titanium oxide with trace amounts of Ru ($Sr_{0.92}Y_{0.08}Ti_{0.98}Ru_{0.02}O_{3+/-\delta}$; SYTRu), which, unlike Ru-loaded SYT (Ru/SYT), displayed higher activity and stability for dry reforming over a wider temperature range [102]. This was attributed to the enhanced oxygen mobility due to the structural transformation in the presence of Ru [102].

Vanadates

Vanadates are less explored anode materials for SOFC. Out of the 29 articles related to vanadates as anode materials for SOFC, four articles are devoted to testing with hydrocarbons [103–106]. Lanthanum strontium chromium vanadates (LSCV) displayed high anode polarization loss [107], which indicates poor catalytic activity. The addition of Pd and GDC to LSCV enhanced the performance by two-fold [108]. In the 450–550 °C range, the TEC showed a sharp bend, suggesting a phase transition in this temperature range. The TECs were $9.6 \times 10^{-6}$ and $11.5 \times 10^{-6}$ °$C^{-1}$ in the temperature ranges of 200–450 and 550–950 °C, sequentially, that are near to that of YSZ [108,109]. As noticed in LSCV, this transformation always had an unfavorable effect on the performance of the anode. A solid oxide fuel cell with a $La_{0.7}Sr_{0.3}VO_3$ anode tested on 5%$H_2S$/95%$CH_4$ fuel mixture

exhibited acceptable cell performance (280 mWcm$^{-2}$) and generated high-value chemicals such as elemental sulfur and CS$_2$ [106]. Thus, La$_{0.7}$Sr$_{0.3}$VO$_3$ anode-based SOFC could be a propitious solution for the processing of sour natural gas.

Molybdates

Smith et al. [110] reported good electrical conductivity for strontium molybdates (SMO)-YSZ. However, it exhibited poor catalytic activity for H$_2$ and CH$_4$ oxidation reactions [110]. Polarization loss was higher than other established perovskite anodes. Sr$_2$FeMoO$_{6-\delta}$ exhibited significant methane oxidation catalytic activity. In contrast, Ca and Ba doped ferrous molybdates were found to display much poorer characteristics [111]. The improved characteristics of Sr$_2$FeMoO$_{6-\delta}$ are assumed to be due to the oxygen vacancies in the crystal structure [111]. Li et al. [112] fabricated a single-cell SOFC containing a Sr$_2$Fe$_{1.5}$Mo$_{0.5}$O$_{6-\delta}$ anode that displayed a power output of 391 mWcm$^{-2}$ at 800 °C utilizing methanol. Further, Yang et al. [113] developed an anode exhibiting many active oxygen species and enhanced oxygen loss by doping Sr$_2$FeMoO$_{6-\delta}$ with lanthanum (Sr$_{2-x}$La$_x$FeMoO$_{6-\delta}$, with x = 0.2). The electrolyte-supported SOFC fabricated with a Sr$_{1.8}$La$_{0.2}$FeMoO$_{6-\delta}$ anode exhibited improved electrochemical performance (885 mWcm$^{-2}$) compared to SOFCs fabricated with Sr$_2$FeMoO$_{6-\delta}$ (740 mWcm$^{-2}$) and Ni–YSZ anodes tested with wet H$_2$ at 800 °C [93]. Additionally, Wang et al. [114] demonstrated enhanced electrical conductivity (16.8 to 26.6 Scm$^{-1}$ at 850 °C in H$_2$) by doping Sm in Sr$_2$Fe$_{1.5}$Mo$_{0.5}$O$_{6-\delta}$. Because of the increased conductivity of the anode, the peak power density of the single cell enhanced from 617 (Sr$_2$Fe$_{1.5}$Mo$_{0.5}$O$_{6-\delta}$ anode) to 742 mWcm$^{-2}$ (Sr$_{1.8}$Sm$_{0.2}$Fe$_{1.5}$Mo$_{0.5}$O$_{6-\delta}$ anode) at 850 °C.

Sr$_2$MgMoO$_{6-\delta}$ (SMMO) has gained considerable attention due to its MIEC, high power density in H$_2$/CH$_4$ fuels, and prolonged stability in H$_2$S (50 ppm)/H$_2$ fuel [115,116]. Additionally, Sr$_2$Mg$_{1-x}$Mn$_x$MoO$_{6-\delta}$ is reported to be a propitious catalyst for CH$_4$ oxidation [117]. Huang and coworkers [115] reported a maximum output of 838 and 440 mWcm$^{-2}$, respectively, in H$_2$ and for CH$_4$ at 800 °C for LSGM-based ESC fabricated from SMMO anode.

The SMMO oxides sintered under reducing atmosphere (5% H$_2$-Ar) exhibit higher conductivity (0.8 Scm$^{-1}$) than those obtained under air (3 × 10$^{-3}$ Scm$^{-1}$) at 800 °C, which is attributed to the electron hopping mechanism due to the reduction of Mo [118]. However, secondary phase formation was observed in the typical operating condition of SOFC [119], which is not good for SOFC.

The most favorable composite catalytic material for the partial oxidation of natural gas was found to be Sr$_2$Ni$_{0.75}$Mg$_{0.25}$MoO$_{6-\delta}$ double perovskite + 50 wt% NiO or 20 wt% SrMoO$_4$ composition because of its ~100% conversion of natural gas compared to other compounds [120]. A large number of double-perovskites-based molybdates have been explored as the anode for SOFC [121–124]. Osinkin et al. [125] explored combustion synthesized Ca-doped double perovskite Sr$_{2-x}$Ca$_x$MgMoO$_{6-\delta}$ with x = 0, 0.25, and 0.5 as a prospective anode for SOFC. The review by Skutina et al. [126] provides an overview of the natural properties of Sr$_2$MMoO$_{6-\delta}$ that facilitate the designing of new generation double perovskite molybdate derivatives for energy conversion and electrochemical purposes.

NiO was easily doped into the B-site of SrV$_{0.5}$Mo$_{0.5}$O$_{4-\delta}$ oxide during the citrate nitrate sol−gel process to synthesize a B-site abundant material SrV$_{0.5}$Mo$_{0.5}$Ni$_{0.1}$O$_{4-\delta}$ [127]. The exsolved nickel nanoparticles (40−140 nm) significantly improved the catalytic activity for the electrochemical oxidation reaction [127]. The Ni–SVM anode displayed outstanding catalytic activity towards H$_2$S-containing fuels and hydrocarbon fuels. The outstanding electrocatalytic activity and stability suggest that Ni–SVM is an important SOFC anode material for various fuels. Another promising molybdate-based anode material studied is Sr$_{2-x}$Ca$_x$Fe$_{1.5}$Mo$_{0.5}$O$_{6-\delta}$, [128]. The study by Istomin et al. [129] showed that Pr$_5$Mo$_3$O$_{16+\delta}$ is of great interest for use as a medium-temperature SOFCs anode material due to its thermomechanical and electrical properties and along with its chemical stability with GDC and YSZ electrolytes. A symmetrical cell was fabricated with combustion synthesized

60 wt% $Sr_2Fe_{1.5}Mo_{0.5}O_{6-\delta}$ (SFM) powder and 40 wt% commercial $Ce_{0.9}Gd_{0.1}O_{1.95}$ on commercial LSGM electrolyte. and detailed electrochemical impedance spectroscopy studies were carried out, and the study revealed higher polarization resistance on the anode side compared to the cathode side [130]. Yang et al. [131] employed solid-state synthesized co-substituted $Sr_2Fe_{1.5}Mo_{0.5}O_{6-\delta}$ ($Sr_2Fe_{1.3}Co_{0.2}Mo_{0.5}O_{6-\delta}$) double perovskite as the anode, and a cell was made with LSGM and LSCF as the electrolyte and the cathode, respectively. The developed anode is promising, as there was no carbon deposition, and the cell exhibited insignificant degradation for 190 h in syngas and 300 h in $CH_4$ fuels (Figure 10).

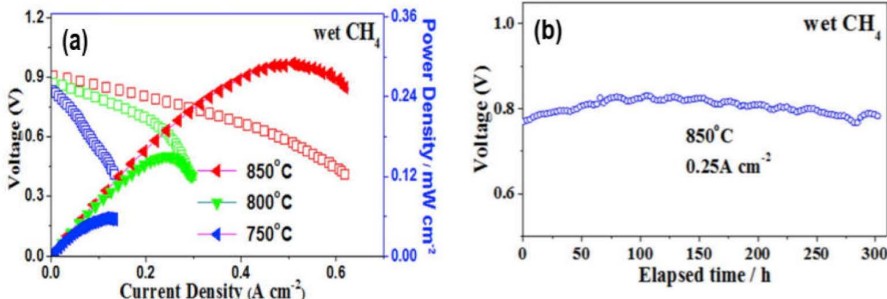

**Figure 10.** (**a**) Voltage–current curves and (**b**) stability test for the SFCM/LSGM/LSCF SOFC in wet $CH_4$ (adapted from [131]).

The use of porous $Y_{0.16}Zr_{0.84}O_{2-\delta}$ (yttria-stabilized zirconia) backbone infiltrated with $Sr_2Fe_{1.5}Mo_{0.5}O_{6-\delta}$ (SFM) was also demonstrated as a promising cathode and anode material for reversible SOFCs [132].

Manganites

Lanthanum strontium chromium manganite (LSCM) is the most popular oxide anode used for HC fuel [133]. Since most of the fuel-cracking reactions take place at above 850 °C, LSCM is found to perform better at higher temperatures [134,135]. The conductivity of LSCM is found to be as low as 0.095 $Scm^{-1}$ [136], which is a major constraint for its usage. A conductivity of 35 $Scm^{-1}$ at 900 °C in air atmosphere and 1 $Scm^{-1}$ in $H_2$ atm was observed for $Ce_xLa_{0.75-x}Sr_{0.25}Cr_{0.5}Mn_{0.5}O_3$. The TEC was in the range of $11.5 \times 10^{-6}$ °$C^{-1}$. However, it reacts with the YSZ electrolyte [137]. The thermal expansion coefficient (TEC) of $(La_{0.8}Sr_{0.2})_{1-y}Al_{1-x}Mn_xO_{3-\delta}$ (LSAM) matches very well with YSZ, and it does not chemically react with YSZ up to 1400 °C. Additionally, the LSAM–YSZ composite anode exhibited good electrochemical performance. However, when the atmosphere was changed from air to wet Ar/4% $H_2$, a reasonably large chemical expansion (0.3–0.5%) was noticed, which resulted in unendurable stress on the thin-film electrolyte layer of a large-area ASC [138]. On the other hand, the poor catalytic activity of $La_{0.8}Sr_{0.2}Sc_xMn_{1-x}O_{3-\delta}$ (LSSM) for methane oxidation brought down the OCV of an LSSM anode-based single cell to 0.55 V [139].

Manganates

Manganates in the form of A-site-ordered double perovskite, $AA'B_2O_{5+\delta}$, have been contemplated as a prospective SOFC anode material [140,141]. $AA'B_2O_{5+\delta}$ is termed as layered perovskite structure. as it follows the stacking sequence $[AO]$-$[BO_2]$-$[A'O_\delta]$-$[BO_2]$-$[AO]$ along the c axis and possesses a large number of vacant $O_3$ sites in the $[AO_x]$ crystal planes. $PrBaMn_2O_{5+\delta}$ (PBMO) is an important electrode material explored for symmetrical SOFC [142]. Felli et al. [142] studied the phase transition between the $Pr_{0.5}Ba_{0.5}MnO_{3-\delta}$ (*m*-PBM) and the double layered perovskite $PrBaMn_2O_{5+\delta}$ (*l*-PBM) and deliberated in the perspective to fabricate versatile electrodes for solid oxide electrolysis cell (SOEC) and SOFC applications. Chen et al. [143] developed epitaxially grown single-crystalline PBMO films on $LaAlO_3$ and studied the nature of surface exchange kinetics and the

oxygen vacancy evolution process. Based on the symmetrical cell study, PBMO was recommended as a good candidate material for SOFCs and chemical sensors due to its highly efficient processes of surface oxygen exchange and excellent stability. Since PBMO suffers from limited electrocatalytic activity, Gu et al. [144] have applied a layer of nano $Pr_6O_{11}$, which decreased the polarization resistance of the cathode and anode and thereby improved its performance. The SOFC cell with $PrBaMn_2O_{5+\delta}$ (PBCMO) anode exhibited higher power densities of 1.77, 1.32, and 0.57 $Wcm^{-2}$ in humidified $H_2$, $C_3H_8$, and $CH_4$, respectively, at 850 °C [144]. The oxygen vacancy concentration of $Pr_{0.5}Ba_{0.5}MnO_{3-\delta}$ further increased with the addition of Mo dopant [145]. This improved the catalytic activity for fuel oxidation [145]. It had helped in bringing down the ASR by 23% (0.13–0.1 $\Omega$ $cm^2$) in $H_2$ and 27% (0.14–0.11 $\Omega$ $cm^2$) in $CH_4$ at 850 °C, and, in turn, the power density raised from 500–600 $mWcm^{-2}$ in $CH_4$ fuel, respectively.

Similarly, Choi et al. [146] improved the electrical conductivity of the PBCMO anode through Ca doping. The oxide displayed electrical conductivity of 48 and 13.4 $Scm^{-1}$ in $H_2$ and air, respectively, at 800 °C. The SOFC cell with symmetrical PBCMO electrodes showed outstanding power density of 1.101, 0.74, and 0.47 $Wcm^{-2}$ at 800 °C in humidified $H_2$, $C_3H_8$, and $C_8H_{18}$ fuel, respectively.

In recent years, there has been a lot of interest in using PBM-based electrodes for reversible SOFC. Kwon et al. [147] found improved electrochemical performance in LSGM composite electrode scaffold infiltrated with PBMO. Tungsten-doped double perovskite, $(PrBa)_{0.95}(Fe_{0.95}W_{0.05})_2O_{5+\delta}$, when used in a symmetrical cell exhibited maximum power densities of 0.610, 0.624, and 0.448 $W$ $cm^{-2}$ at 800 °C for syngas, ethane, and propane, respectively [148].

Ni-doped perovskites, $(Pr_{0.5}Ba_{0.5})_{1-x/2}Mn_{1-x/2}Ni_{x/2}O_{3-\delta}$ with x = 0, 0.05, 0.1, and 0.2 (S-PBMNx)), were prepared to design exsolution systems as solid oxide fuel cell anodes and for catalysis applications. The anode with the highest density of exsolved particles showed the best electrochemical performance [149]. Exsolved $Ni/LaSrMnO_{4\pm\delta}$ was highly resistant to carbon formation on the Ni surface [150]. However, it is irreversibly poisoned with 50 ppm of $H_2S$ at 850 °C.

### 4.1.2. Fluorite Structure

Fluorites are known to possess lattice symmetry, which benefits ionic conduction. The DMO at IT was influenced by concentration polarization at low potentials, ensuing from a higher oxidation rate at the catalyst surface compared to the rate of $O^{2-}$ feed at the TPB. Hence, it is essential to have anodes with high ionic conductivity. Some of the promising catalysts possessing fluorite structure are discussed below.

### Ceria-Based Oxides

Ceria is a propitious catalyst for anode catalysis as it is not as active as Ni and displays higher resistance to carburization. Doped ceria displays higher ionic conductivity [151,152]. Since ceria electrochemically oxidizes the deposited carbon, the performance of the anode is not affected due to HC cracking reactions [151]. In fact, traces of carbon deposition facilitated electronic conduction to some extent [153]. Ramirez et al. [154] suggested that gadolinia-doped ceria (GDC) tended to possess a reaction mechanism in which the rate is checked by the slow reaction between the adsorbed methane/surface HC species and $O_2$ in GDC. Subsequently, an easy reaction between steam and GDC replenished $O_2$ in the reaction site. GDC is highly resistant to carburization in steam reforming conditions even for a low steam/methane ratio of 0.6. Additionally, it is worth noting that $H_2$ played a prominent inhibitory effect on the rate of reforming. Niobium-doped $CeO_2$ had a higher catalytic activity than $CeO_2$ due to the controlled grain growth [155]. SOFC with Mn-doped ceria–ScSZ as anode exhibited a reasonable performance of about 262 $mWcm^{-2}$ in the $CH_4$ fuel at 900 °C [156]. Since the anode possesses poor catalytic activity for C–H scission, the cell has to be operated at high temperatures. Hence ceria is mostly preferred in cermets rather than as a ceramic oxide anode.

Apart from these, ceria–zirconia solid solutions such as $Ce_{0.1}Zr_{0.9}O_2$ (CZO) have been used as an SOFC anode material. The CZO was found to show lower area-specific resistance (ASR). Additionally, it displayed reasonable methane conversion for combustion reaction [157,158]. However, the electronic conductivity remained substantially lower than cermet anodes.

### 4.1.3. Other Oxides

Other oxides with pyrochlore [159], spinel [160], rutile [161], and tungsten bronze [162] structures have also been evaluated as SOFC anodes. However, most of these oxides suffer from low ionic conductivity, low CTE, poor thermal stability, and poor catalytic activity. Overall, the above ceramic oxides have displayed low to moderate performance except in a few exceptional cases such as $Ce(Mn,Fe)O_2$, etc. [163]. Often, these oxides were developed for an ESC configuration and tested in a similar configuration. Even the test procedure can play a role in the performance of the electrode. Since oxides can overcome some of the limitations of Ni–YSZ cermet anodes such as metal coarsening, carburization, etc., they are being used in combination with Ni–YSZ cermet anodes. Runge et al. [164] have studied $U_{1-x}M_xO_{2-\delta}$ (M = Mg, Ca, Sr) and found higher electrical conductivity in an anode working environment. The Ca-doped $UO_2$ ($U_{0.823}Ca_{0.177}O_{2-\delta}$) exhibited a high conductivity of about $3\,Scm^{-1}$ at $P_{O2} < 10^{-4}$ atm at 600 °C [164]. However, $U_{1-x}M_xO_{2-d}$ materials are not suitable as anodes for SOFC due to the lower ionic transference number ($t_i = 0.01$).

Oxide anodes are mostly used in electrolyte-supported SOFC. These anodes were mostly of the perovskites and fluorite structures. In order to understand the relative performances of these anodes, the peak power density of different oxide anodes reported at 800 °C is compared. From Figure 11a,b, it is evident that perovskites exhibited the best performance among the oxide anodes in both $H_2$ and methane fuel. However, the performance of anodes was reduced to half in methane fuel.

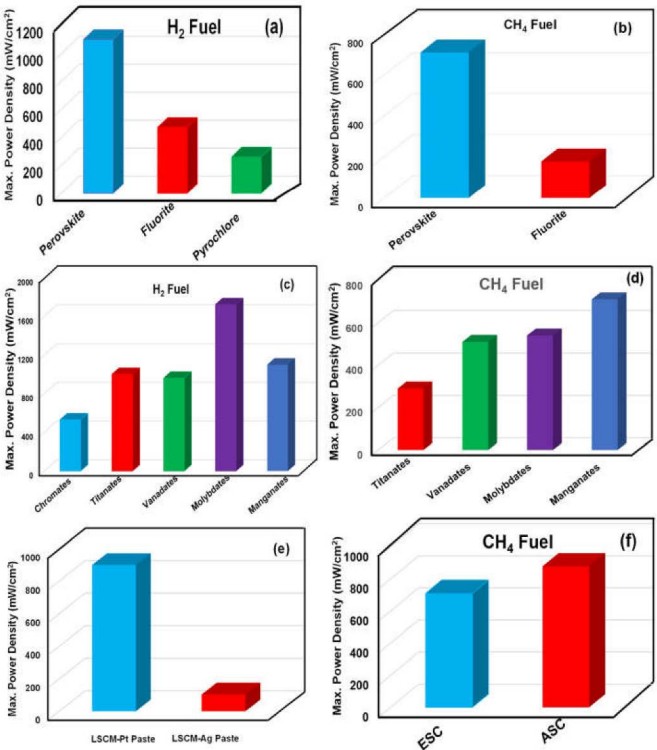

**Figure 11.** Performance of oxide anodes with the different crystal structure (**a**) in $H_2$ and (**b**) $CH_4$; with different composition in (**c**) in $H_2$ and (**d**) $CH_4$; LSCM (**e**) with different current collector pastes and (**f**) different configurations.

Among perovskites, manganates and molybdates showed better performance (Figure 11c,d). LSCM is the most successful composition to date. Ceramic oxides have low electronic conductivity and are hence preferably used in ESC as there is a possibility for the interference of current collection Pt paste, which leads to an overestimation of anode performance. Kim et al. [165] emphasized the function of the Pt contact layer in enhancing the performance of the oxide anodes. Particularly, in the case of LSCM, there was about a five-fold improvement in performance while using Pt paste as the current collector (Figure 11e). Therefore, higher performance was reported for oxide anodes-based ESC over ASC (Figure 11f). To improve the catalytic activity, noble metals and rare-earth-doped ceria were infiltrated into the oxide anodes.

## 4.2. Cermets

Cermets are porous composite structures of metal and ceramics. Among metals, Ni, Co, Pd, Rh, and Ru, show excellent catalytic oxidation activity [36,41]. Accordingly, inexpensive Ni and Co are mostly preferred for the cermet. However, SOFC electrodes should possess mixed ionic electronic conductivity (MIEC) to catalyze the electrochemical reaction. Therefore, $O_2$ ion-conducting ceramic materials such as YSZ and ceria are added to the cermet. Additionally, this will be helpful in matching the TEC to the electrolyte and controlling the metal sintering.

### 4.2.1. Ni-Based Cermet

Currently, Ni–YSZ is the popular anode system that exhibits all required properties for SOFC such as electronic/ionic conductivity, catalytic activity, thermal stability, chemical compatibility, TEC, etc. Several investigations have been carried out to optimize Ni–YSZ to accomplish high electrical conductivity, to tailor both TECs, and to evade the seclusion of Ni-particles [166–169]. The best electrochemical performances are obtained with 20–40% porous Ni–YSZ with 40 vol

Koh et al. [172] demonstrated the working of Ni–YSZ-based ASC in $CH_4$. By maintaining the carbon activity below one, i.e., a thermodynamically carbon-free condition, the functioning of SOFC was demonstrated up to 2000 h, and no major performance degradation was observed. Further, they reported the reversible nature of carbon deposits for the cell operated in presence of steam. However, dry $CH_4$ fuel created irreversible damage to the performance. Eguchi et al. [173] demonstrated internal reforming of $CH_4$ over Ni–YSZ electrode layers but, to achieve complete conversion, an adequate amount of anode catalyst was required. The reactivity of the catalyst for internal reforming or fuel conversion should be considered along with the electrochemical oxidation while operating with multi-component carbon-based fuels. Further, the I–V characteristics realized for the CO–$CO_2$ fuel systems were invariably inferior to those obtained for the $H_2$–$H_2O$ fuel system.

Thus, it is evident that Ni–YSZ cermet has to be modified to accommodate the HC fuel. Though the thermodynamically carbon-free condition alleviates the carburization, it cannot completely control the carbon deposition. The modification of Ni–YSZ should improve the catalysis of reforming reaction and ensure the complete utilization of HC fuel. Accordingly, there were efforts to modify the Ni–YSZ cermet. Wang et al. [174] synthesized Ni and Ru bimetal-doped perovskite catalyst, $Ba(Zr_{0.1}Ce_{0.7}Y_{0.1}Yb_{0.1})_{0.9}Ni_{0.05}Ru_{0.05}O_{3-\delta}$ (BZCYYbNRu), and, when applied as a steam reforming catalyst layer on a Ni–YSZ-supported anode, the single fuel cell, apart from displaying a higher power density of 1113 mW cm$^{-2}$ at 700 °C with a continuous feed stream of 10 mL min$^{-1}$ $C_4H_{10}$ at an $H_2O/C = 0.5$, exhibited a much better operational stability for 100 h at 600 °C. These results are better than those reported in the literature.

The addition of a functional layer can facilitate active sites for internal or external steam reforming for the fuels before passing to the anode. Nevertheless, by the introduction of a reforming layer, delamination happens during thermal cycling due to the incompatibility

between the catalyst and anode layers. Additionally, current collecting is difficult due to the poor electrical conductivity of the catalyst layer.

Sumi et al. [175] observed dependence of durability under HC fuel on the $O^{2-}$ ion conductors in the Ni-based anodes. The Ni–ScSZ displayed better endurance than Ni–YSZ at 1000 °C in $CH_4$ fuel. Additionally, the electrolyte-supported SOFC with Ni–SSZ showed performance in direct biogas fuel at the operating temperature of 1000 °C without carbon deposition. Ni–GDC showed still better durability because GDC exhibits higher catalytic activity for carbon species oxidation [175,176]. Muccillo et al. [177] studied the Ni–GDC electrode in ethanol fuel and found that doped ceria could not resist carbon formation due to its poor catalytic activity for ethanol conversion. The Ni–GDC anode exhibited high polarization resistance compared to Ni–YSZ in ethanol fuel at 900 °C. The transformation of fluorite-type $CeO_2$ to $Ce_3O_5$ rare-earth C-type structure in the presence of ethanol at a high temperature was reported [177]. The ordering of oxygen vacancies in the C-type structure can reduce the ionic conductivity [178]. This was considered the possible reason for increased polarization. Accordingly, ceria is regarded as a low-temperature (<800 °C) anode catalyst. Qiu et al. [179] modified Ni–GDC anode-supported cells with a $Sr_2Co_{0.4}Fe_{1.2}Mo_{0.4}O_{6-\delta}$ (SCFM) layer outside the anode support. It was observed that the SCFM layer facilitated the enhancement in the electrochemical furnace and durability and efficiently performed dry reforming when $CH_4$–$CO_2$ was used as fuel. Yano et al. [180] operated a single-chamber SOFC with Ni–SDC at temperatures as low as 300 °C in ethanol, and the SOFC exhibited an output of 44 mWcm$^{-2}$. Though the lower operating temperature is appealing, it will lead to other issues such as partial oxidation behavior in the case of HC fuel [181]. Additionally, there were attempts to use Ni-$Y_{0.25}Zr_{0.60}Ti_{0.15}O_{2-x}$ (YZT). However, the lower ionic conductivity of YZT reduced the electrochemical performance of the anode [182].

The maximum performances of SOFCs with Ni-based cermet anodes in both ASC and ESC configuration claimed in the literature are displayed in Figure 12. In general, the cermet anodes with zirconia operated at relatively high temperatures (100–150 °C higher) than anodes with ceria. To understand the relative performance, the best performance of different Ni–cermet anodes is compared. The Ni-rich anodes showed better thermal stability and are widely used in ASC. However, Ni–YSZ anodes fail within a few hours of operation in dry $CH_4$ fuel if proper precautions are not taken. Therefore, Ni is used in combination with metals such as Cu, Co, or Fe. Due to the adjustment of atomic orbital and electron cloud effectively, the coking resistance increases. Bi-metallic Ni-based cermets are discussed in later sections. Additionally, the ionic conductivity of cermets plays a vital role in controlling carburization. Oxygen ion conductivity of cermet should be high to improve the durability under HC fuel. Accordingly, ceramic oxides such as GDC and SDC have been used to electrochemically oxidize the deposited carbon. The best performances are obtained with a porosity of 20–40% and with 40 vol

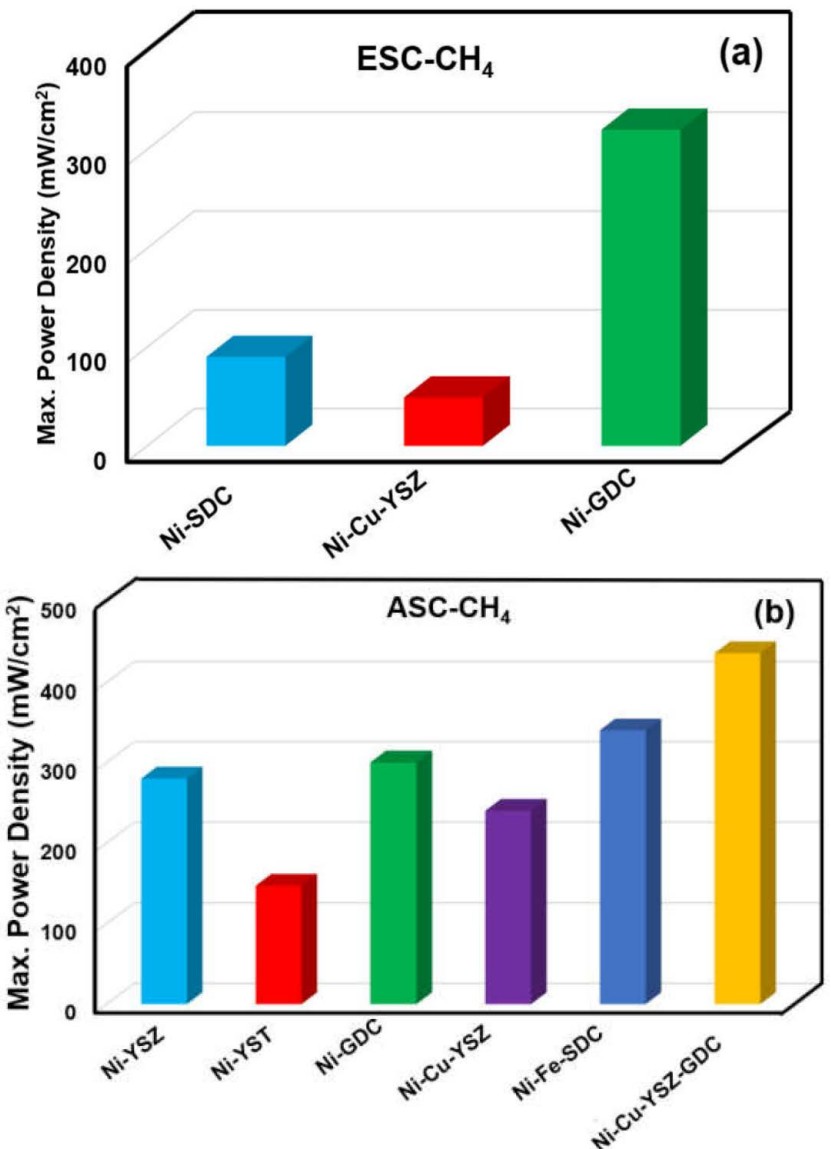

**Figure 12.** Performance of Ni cermet anodes in (**a**) ESC and (**b**) ASC operated with CH$_4$.

### 4.2.2. Cu-Based Cermets

Though Cu is recognized for curtailing carbon deposition, its catalytic activity is poor for electrochemical oxidation of HC fuels. Hence, Cu requires a supportive catalyst in the anode. Often Cu has been used in combination with good oxidation catalysts such as ceria. The Cu–CeO$_2$–YSZ system offered an opportunity to study the merits of oxygen ionic conductivity and catalytic oxidation activity of CeO$_2$. The improved performance of Cu–CeO$_2$–YSZ over Cu–YSZ highlights the catalytic activity in addition to charge transfer functionality in the anode. Although the stability of Cu–YSZ in 5000 ppm sulfur was impressive, the low catalytic activity of Cu–YSZ reduced the OCV at 700 °C [183]. Cu–CeO$_2$ displayed excellent catalytic activity for the electro-oxidation of combustible species prevailing in the anodic active electrochemical zone and carburization reduced when oxide ions were provided electrochemically to the catalyst [184]. The 21.5%Cu–8.5%CeO$_2$–SSZ cells prepared by infiltration showed power density up to 438 mWcm$^{-2}$ in ethanol [185]. However, 40%Cu–20%CeO$_2$–YSZ displayed partial oxidation behavior in n–butane fuel [181]. Additionally, Ramirez-Cabrera et al. [186] showed that in GDC the reaction rate was controlled by a slow methane adsorption process. All these results emphasize the requirement of additional support for electrochemical oxidation of HC even

in the Cu–ceria system. Lu et al. [187] obtained enhanced performance by doping 1.5%Pd in the Cu–LSCM anode. Akdeniz et al. [188] studied Cu and $CeO_2$ infiltrated Ni-based SOFC anodes in dry methane fuel, and the cell exhibited a maximum power density of 250 mWcm$^{-2}$ at 700 °C, which degraded after 6 h.

The performances of SOFCs with Ni-related cermet anodes in both ASC and ESC configurations claimed in the literature are shown in Figure 13. In general, the cermet anodes with zirconia operated at relatively higher temperatures (100–150 °C higher) than anodes with ceria. To understand the relative performance, the best performance of different Ni–cermet anodes is compared. The Ni-rich anodes showed better thermal stability and are widely used in ASC. Without suitable measures, Ni–YSZ anodes fail within 25 h of operation in dry $CH_4$ fuel. Therefore, Ni is used in combination with other metals as discussed in Section 4.2.1. Bi-metallic Ni-based cermets are discussed in later sections. Additionally, the ionic conductivity of cermets plays a vital role in controlling carburization. Oxygen ion conductivity of cermet should be high to improve the durability under HC fuel. Accordingly, ceramic oxides such as GDC and SDC have been used to electrochemically oxidize the deposited carbon. Best performances are obtained with a porosity of 20–40% and with 40 vol

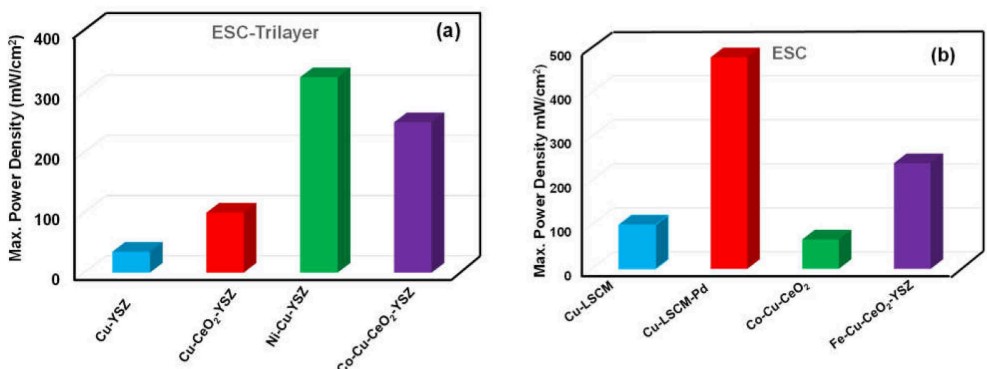

**Figure 13.** Bar graph showing the maximum power density of SOFCs fabricated from Cu-cermet anodes (**a**) using single, bi-layer, and tri-layer infiltration and (**b**) in ESC configuration.

### 4.2.3. Other Metal-Based Cermets

Polarization losses in Co–YSZ are higher when compared to Ni–YSZ [189]. Additionally, Co is susceptible to carburization, and Ru and Ir are low in TEC and expensive, whereas Ru is extremely costly [190]. Hence, a simple composite, i.e., a single metal in combination with YSZ, is difficult to realize and, hence, an alternative way is to use bimetallic cermet. Iron addition decreases the nickel catalytic activity for C–H scission and adjusts the TEC. In the same way, copper addition controls carbon deposition [191,192]. Therefore, bimetallic anodes were preferred to negate the issues associated with HC fuel.

### 4.2.4. Ni-Cerate/Zirconate-Based Cermets

The development of electrodes based on barium cerate or barium zirconate may provide solutions to many SOFC problems. Doped barium cerate and barium zirconate are known for their high ionic conductivity in the IT range, and they possess the capability to carry both proton and oxygen ion vacancies. The review article by Kasyanova et al. [193] presents an overview on the proton-conducting electrolytes of $BaCeO_3$, $BaZrO_3$, or $BaCeO_3$-$BaZrO_3$ families (designated as $Ba(Ce,Zr)O_3$) and on the design of electrodes with the same cations at A- and B-positions of the $ABO_3$ perovskite structure for proton-conducting fuel cells (PCFCs).

A large number of groups have explored the application of $Ba(Ce,Zr)O_3$-based electrodes for protonic ceramic electrochemical cells [194–198]. The Ni–$BaZr_{0.1}Ce_{0.7}Y_{0.1}Yb_{0.1}O_{3-\delta}$ (BZCYYb) anode is proven for its coking tolerance [199]. A single cell with Ni–BZCYYb, SDC, and LSCF, respectively, as the anode, the electrolyte, and the cathode displayed a sta-

ble output of 600 mWcm$^{-2}$ in dry propane for 24 h, and the performance durability of cells with Ni-GDC, Ni-GDC+BCY, and Ni-GDC+BCYb measured at 200 mAcm$^{-2}$ and 750 °C in wet CH$_4$ as fuel and air as the oxidant showed the best performance for Ni–GDC+BCYb. Ni-based bimetallic alloy catalysts attached to BaZr$_{0.4}$Ce$_{0.4}$Y$_{0.1}$Yb$_{0.1}$O$_{3-\delta}$ (BZCYYb) can be directly applied to the PCFC anode for internal steam reforming of methane at a low temperature [200]. The composition of 6Ni2Rh showed the highest catalytic activity, as evident from the temperature-programmed reduction studies, at all temperatures and, hence, it can be a propitious reforming catalyst in PCFC anodes [200]. Nishikawa et al. [201] have explored double-layer hydrogen electrodes with catalyst layers of BaCe$_{0.50}$Zr$_{0.27}$ Y$_{0.20}$-Ni$_{0.03}$O$_{3-\delta}$ (BCZYN) and Ni on a BaCe$_{0.10}$ Zr$_{0.70}$Y$_{0.20}$O$_3$ electrolyte and current-collecting layers (CCLs) for using them in PCFCs. The BaZr$_{0.85}$Y$_{0.15}$O$_{3-\delta}$–NiO (BZY15-NiO) cathode and the BaZr$_{0.85}$Y$_{0.15}$O$_{3-\delta}$ (BZY15) electrolyte were applied by pulsed laser deposition (PLD) on metal supports for PCFCs at 700 and 600 °C, respectively [202]. However, cell performance in a hydrocarbon environment is not studied. The catalytic activity of Ni–Ba(Zr,Y)O$_{3-\delta}$ (BZY) cermets for ammonia decomposition has been explored and Ni–BZY showed higher activity than Ni–YSZ due to the high basicity of BZY and the high resistance to the hydrogen poisoning effect. The electrochemical performance studies were also carried out for the ASC of Ni–BZY/BZY/Pt with separate NH$_3$ and H$_2$ as fuels [203].

*4.3. Bimetallic Cermets*

4.3.1. Ni–Cu Systems

Many researchers validated the suppression of carburization when Cu was added to the anode [204,205]. Copper is a perfect element for alloying Ni to circumvent carburization due to its inferior catalytic activity towards C–C bond formation and cleavage of C–H and C–C bonds. Additionally, Cu melts at 1083 °C and Ni melts at 1453 °C. Therefore, it is essential to reduce the Cu quantity in the alloying composition. The melting point of Ni–Cu alloy increases linearly with Ni content because of the formation of isomorphous substitutional solid solution. Due to the alloy formation, the catalytic activity of Ni for alkane hydrogenolysis and dehydrogenation reactions of anode is altered [206–208].

Lu and co-workers [209] studied Ni$_{0.7}$Cu$_{0.3}$–YSZ/Ni$_{0.3}$Cu$_{0.7}$–YSZ systems and reported an open circuit voltage of SOFC near to the theoretical value, but there was a very little drop in the output current and power density while altering the fuel from H$_2$ to coal gas. The anode responded rapidly to the change-over of fuels without obvious delay. Further, the anode displayed self-cleaning of coke deposition [209]. Notable enhancement in power density after 500 h operation in dry methane for SOFC fabricated from the Cu80%–Ni20%–YSZ anode has been reported [210]. The impedance spectra with similar fuel cells showed the formation of few carbon deposits with time, and the output increased with enhanced electronic conductivity of the anode.

Woo and coworkers [211] electroplated Cu on Ni–YSZ using an aqueous copper sulfate bath to fabricate Cu–Ni–YSZ. The Cu–Ni–YSZ ASC displayed stable performance up to 200 h. On the contrary, Ni–YSZ ASC degraded steeply within 21 h due to carbon deposition [211]. The CuO addition enhanced the sinteractivity of the NiO–SSZ anode [212]. The ohmic resistance of the anode was reduced with CuO addition due to enhanced electronic conductivity. Similarly, 6 wt% Cu-doped Ni–YSZ showed improved carbon resistance [213]. Kumar et al. [214] reported a high-performance (436 mWcm$^{-2}$ at 850 °C in CH$_4$) ASC fabricated from Ni$_{0.9}$–Cu$_{0.1}$–YSZ$_{0.95}$–GDC$_{0.05}$ anode whose performance was attributed to the intact anode–electrolyte interface resulting due to co-tapecasting followed by co-firing. This study demonstrated the prospects of utilizing Cu in a high-temperature fabrication process to develop a HC-compatible SOFC with improved performance [214]. A 2 mm thick electrolyte-supported single cell (ESC) with a Ni$_{0.9}$Cu$_{0.1}$YSZ$_{0.95}$GDC$_{0.05}$ anode was also fabricated. From Figure 14, it is evident that the performance of the cell in H$_2$ and CH$_4$ were almost similar above 800 °C. However, the performance difference was magnified with reducing temperature. For instance, at 767 °C, the performance of the cell in H$_2$ fuel was about 40 mWcm$^{-2}$, whereas in CH$_4$ fuel, it was around 33 mWcm$^{-2}$ [96].

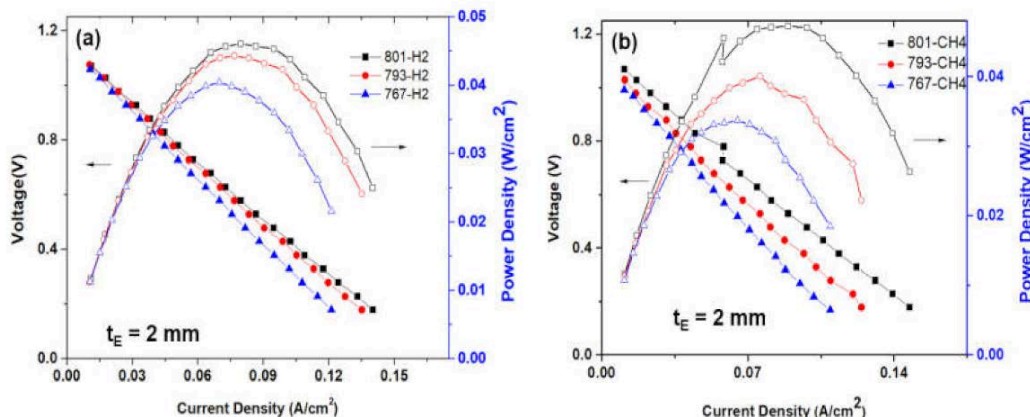

**Figure 14.** Performance of 2 mm electrolyte-supported SOFC with $Ni_{0.90}Cu_{0.10}YSZ_{0.95}GDC_{0.05}$ anode in (**a**) $H_2$ fuel and (**b**) $CH_4$ fuel at ■ 801 °C, ● 793 °K, and ▲ 767 °C; filled shapes correspond to the V–I curve (adapted from [96]).

To identify the factors responsible for the deteriorating performance at lower temperatures, impedance analysis was carried out at 0.7 V in the temperature range of 801–767 °C. Impedance spectra revealed that the magnitude of polarization resistance of the electrode remains almost the same with decreasing temperature. As expected, the electrolyte ohmic resistance increased drastically at lower temperatures. Based on these facts, the difference in performance has been attributed to the poor oxidation of methane at lower temperatures owing to the reduction in oxygen ionic conductivity across the electrolyte. The slope of I–V for methane fuel was steeper than $H_2$. Further, the OCV for $H_2$ fuel remained almost constant in the investigated temperature range, whereas, for methane, OCV gradually decreased with decreasing temperature. These results suggested different Faradic oxidation for methane and $H_2$. This validates the indirect oxidation reaction pathway discussed by Mogensen et al. [62].

### 4.3.2. Ni–Fe System

Ni–Fe alloy is an excellent system for IT SOFC with $H_2$ fuel. It displayed a peak power density of 1333 mWcm$^{-2}$ in $H_2$ fuel at 650 °C [215]. Kan et al. [216] studied this system in dry methane fuel and observed a good reduction in carbon deposition. However, the maximum power density was 340 mWcm$^{-2}$ at 650 °C, which was four-fold lower than in $H_2$ [216]. Additionally, the temperature-programmed reduction studies indicated higher reduction temperatures for Fe and Fe–Ni alloy. The poor reduction efficiency can eventually lead to poor anode catalysis [217]. Eventually, these results would lead to the question of whether IT-SOFC is advantageous for HC fuel. The fact is that the partial oxidation behavior of HC and the eventual choking of the fuel channel are serious issues that outweigh the benefits of IT-SOFC.

Though the Ni–Fe system in $H_2$ displayed a very high performance at 650 °C, this does not guarantee better performance at high temperatures. For instance, Ni–Fe–YSZ displayed marginal performance at 800 °C [218] because the reaction environment at low temperature is completely different from the high temperature. At low temperatures, carbon activity plays a more major role than carbon diffusivity in metal. Therefore, just the suppression of the Boudouard reaction can reduce the carbon activity an,d in turn, the carburization can come down. As the Fe does not catalyze this reaction, the carburization is reduced at 650 °C, whereas, in the case of high temperature, the Boudouard reaction is curbed, and the carbon diffusion in the metal increases the severity of carburization. Hence, Ni has the advantage over Ni–Fe in the perspective of carbon diffusivity.

### 4.3.3. Co–Cu System

The Co–Cu system is one of the potential bimetallic anode cermets for HC fuel. Co has many properties similar to Ni. Among the metals known to possess good electrocatalytic

activity, only Ni and Co are relatively inexpensive [219,220]. Though cobalt melts at moderately higher temperatures compared to nickel, it can be reduced under identical conditions as that of nickel. Unfortunately, Co is also prone to carburization, and carbon fibers are formed in dry methane. In contrast to Ni, Co has insubstantial solubility with the Cu at SOFC operating temperatures. Furthermore, Cu tends to seclude on the surface in the process of minimizing surface energy. Lee et al. [221] co-impregnated copper and cobalt salts in porous YSZ and subsequently reduced it to develop a Cu–Co-based anode. The 50:50 Co–Cu cermet with 15 wt% $CeO_2$, 30 wt% metal, and rest YSZ showed stable performance in n–butane up to 500 h and displayed a power density of 250 mWcm$^{-2}$ in $CH_4$ fuel at 800 °C. Due to the sintering of the Cu phase, there will be a discontinuity in the Co conductive network of the electrode, which will lead to loss of conductivity of the entire anode composite. Accordingly, to ensure the continuity of the conductive Co phase network, Gross et al. [220] electrodeposited Co onto a reduced Cu to fabricate Cu–Co electrodes. The electrodes exhibited improved thermal stability compared to Co–$CeO_2$–YSZ electrodes fabricated by the impregnation of higher loading of cobalt due to the formation of interconnected structures. Since Cu easily diffuses through the Co film and forms a monolayer of Cu on the electrodeposited Co, the Co–Cu–$CeO_2$–YSZ electrodes displayed superior resistance to carbon formation in HC fuels, as evident in Figure 15. Thus, it was opined that Cu migration onto the surface of Co was responsible for the stability of Cu–Co anodes against carburization. It was also believed that the catalytic activity originated from isolated Co atoms is present within the Cu-rich phase [220].

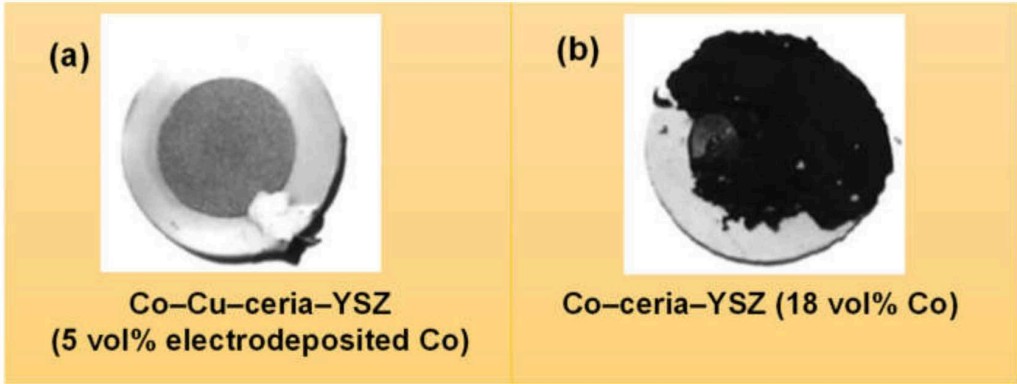

**Figure 15.** Photograph of the cells (**a**) after reduction in dry $H_2$ at 800 °C and (**b**) exposure to dry $CH_4$ for 3 h at 800 °C (adapted from [220]).

Sarruf et al. [222] disclosed the use of the ceria–cobalt–copper anode. The reduction of ceria to $CeO_{2-x}$ was hypothesized, which accounts for oxygen deficiency that, in turn, polarizes the structure and enhances the conductivity. The oxygen storage capacity (OSC) of ceria will be enhanced depending on the ceria's particle size and the oxygen under-stoichiometry over the surface of the ceria [222]. The developed cell was stable in HC fuel. The SOFC single cell showed a power density of 411 and 95 mWcm$^{-2}$, respectively, in $H_2$ and $CH_4$ fuel at 0.7 V and 850 °C.

### 4.3.4. Ni–Co System

Since $Co^{2+}$ has a higher solubility than $Cu^{2+}$ in NiO lattice, Ni–Co alloys are anticipated to exhibit improved electrochemical activities and thermal stabilities, without the aid of any unconventional processing techniques such as infiltration. Further, the oxidation resistance of Co is better than Ni and is not anticipated to display corrosive properties at high overpotentials or high $pO_2$ [189]. It was well-established that, by adding small amounts of cobalt to nickel cermets, electrochemical activation energy and pre-exponential factor can be increased, indicating better dispersion of larger metals [223].

Grgicak et al. [224] obtained an exchange current density ($i_o$) of 51 and 44 mAcm$^{-2}$ in dry $H_2$ and $CH_4/H_2S$, respectively, at 850 °C for the $Ni_{0.92}Co_{0.08}$–YSZ system. Impressively, $Ni_{0.69}Co_{0.31}$–YSZ had $i_o$ of 94 mAcm$^{-2}$, which was higher than $H_2$, which indicated its ability to catalyze the CO oxidation reaction [224]. Brien et al. [225] investigated the influence of $CO:H_2$ fuel ratio on the performance of SOFC. A drastic decrease in exchange current density for 7 days was observed, which was assigned to a change in the microstructure of the anode cermet [225].

### 4.3.5. Fe–Cu System

Kaur et al. [226] explored the usage of Fe as second metal to Cu. The anode composite with a 1 M Cu:1 M Fe ratio had better interconnectivity of the metallic phase. It resulted in improved SOFC power density (240 mWcm$^{-2}$) in methane fuel at 800 °C. As discussed earlier, Fe does not catalyze the Boudouard reaction and, hence, it is advantageous at lower temperatures (<700 °C). Due to relatively higher carbon diffusivity in Fe than Ni at higher temperatures, the tendency of carburization increases.

### 4.3.6. Ni–Mo System

A maximum SOFC performance of 594 mWcm$^{-2}$ in methane fuel using a 3:1 Ni–Mo alloy anode composite has been claimed [227]. The anode-supported cell prepared by the tape casting–infiltration process exhibited stable performance for 120 h due to the presence of Ni-rich intermetallic phases and Ni–lean phases that are anticipated to display coking and sulfur tolerance. The carbon deposition is known to begin with the C−C bond creation on the catalyst surface that requires a minimum of two carbon-activation sites. Hence, it was hypothesized that the Mo atoms possessing lower activity than Ni towards methane activation can predominantly dilute the active sites of Ni, thereby successively concealing C−C bond formation. However, the low TEC of Mo is a major obstacle for the preparation of ASC by the conventional tape co-firing technique. Further, there were reports on the formation of molybdenum carbide from molybdenum oxide using 20% $CH_4$–$H_2$ reducing gas mixture at 700 °C [228]. This may alter the microstructure and eventually the mechanical integrity of the cell.

## 5. Long Term Stability of Hydrocarbon Compatible SOFC Anodes

Though there are a large number of reports on the synthesis of hydrocarbon compatible novel anode compositions, only a few researchers have studied the long-term stability of the anodes in the presence of hydrocarbons, and the data are represented in Table 2 [140,229–237]. Sengodan et al. [140] have carried out stability studies of layered PBMO with Co–Fe catalyst under a constant current load of 0.2 A cm$^{-2}$ at 700 °C in $C_3H_8$ for 500 h. A single-cell SOFC containing 2.5 wt% MgO in-filtrated in Ni–SDC cermet anode showed long-term stability for 330 h operated at 0.8 V at 800 °C with 3% water humidified methane as fuel [230]. The excellent coking tolerance performance of the MgO-modified Ni cermet anode has been associated with the enhanced adsorption property of $H_2O$ and $CO_2$ on MgO.

**Table 2.** Long-term stability data of SOFCs with hydrocarbon compatible anodes.

| Anode | Electrolyte | Cathode | Fuel | Temp. °C | Power Density (Wcm$^{-2}$) | Stability | Ref. |
|---|---|---|---|---|---|---|---|
| (PrBa)$_{0.95}$(Fe$_{0.9}$Nb$_{0.1}$)2O$_{5+\delta}$ oxide | LSGM | PrBaCo$_2$O$_{5+\delta}$ | H$_2$<br>wet CH$_4$<br>dry CH$_4$ | 800 | 1.05<br>0.64<br>0.57 | Highly coking resistant in dry CH$_4$ for 200 h | [229] |
| MgO-modified Ni–SDC | LSGM | SCCO-SDC | Humidified CH$_4$ | 800 | 0.714 | Excellent long-term stability for 330 h | [230] |
| Ni–SDC | SDC | SSC-SDC | Liquid methanol | 650 | 0.698 | No degradation during 60 h long-term testing | [231] |
| Li-modified Ni–SDC<br>Na-modified Ni–SDC | LSGM | SCCO-SDC | Wet CH$_4$ | 800 | 0.212<br>0.231 | Stability up to 70 h | [232] |
| MoO$_2$-based anode | YSZ | LSM | n-dodecane | 850 | Initially 0.34; after optimizing porosity of anode = 2.5 | Highly coke resistant with improved long-term stability | [233] |
| Ni–GDC with 0.5 wt% Sn catalyst | GDC | LSCF-GDC | Dry CH$_4$ | 650 | 0.93 | Operates for over 40 h without degradation 24 h | [234] |
| MoO$_2$ porous thin film deposited on Ni–YSZ | YSZ | LSM | Mixture of n-dodecane and air | 750 | >4.0 | Highly coking resistant | [235] |
| Ni-Fe-LSGM | LSGM | SSC | CH$_4$ | 800 | 0.48 | Good stability as no coke deposition after 15 h | [236] |
| Ni-infiltrated porous GDC scaffolds | GDC | LSCF | Wet CH$_4$ | 600 | 0.125 | Stable operation for 24 h | [237] |
| Layered PrBaMn$_2$O$_{5+\delta}$ anode with PrBaMn$_2$O$_{5+\delta}$ and Co–Fe catalyst | LSGM | NdBa$_{0.5}$Sr$_{0.5}$ Co$_{1.5}$ Fe$_{0.5}$ O$_{5+\delta}$–Ce$_{0.9}$ Gd$_{0.1}$ O$_{2-\delta}$ (NBSCF50-GDC) | C$_3$H$_8$ | 700 | 1.32 | Long-term stability tested for 500 h | [140] |
| Ni–YSZ | YSZ | LSM-YSZ | | | | | |

## 6. H$_2$S Poisoning Issue

Though the main problem associated with HC fuel is the carburization of the anode, all the currently existing HC fuels contain traces of sulfur. In general, SOFCs are proposed to generate power from carbon- and sulfur-containing fuels such as syngas obtained from coal or biomass, natural gas, etc. In any case, the sulfur is present in biogas, diesel, and LPG up to 700, 10, and 50 ppm, respectively. Kishimoto et al. [238] have constructed Ellingham diagrams (oxygen potential vs. temperature) plots to study the possible effects of sulfur on SOFC Ni anodes and analyzed from the thermodynamic considerations of Ni–S–C–O–H systems. Nickel sulfides existing in numerous defined compounds such as NiS, NiS$_2$, Ni$_3$S$_2$, Ni$_3$S$_4$, Ni$_9$S$_8$, Millerite (NiS), heazlewoodite (Ni$_3$S$_2$), polydymite (Ni$_3$S$_4$), and vaesite (NiS$_2$) are the most standard minerals [239]. Ishikura et al. [240] have studied the influence of H$_2$S poisoning on the anode layer of SOFC. The study revealed the formation of nickel sulfides when H$_2$S containing fuel is fed to the Ni–YSZ anode. The melting of nickel sulfides leads to the change in the morphology of the anode structure, resulting in a reduced area of triple phase boundary, plugging of the anode pores, and breaking of the nickel network. In the case of YSZ, the poisoning will happen due to the reaction between yttrium and sulfur. As a result of this reaction, Y$_2$O$_3$ segregation occurs in YSZ, and the conductivity of YSZ decreases.

The adsorption of sulfur on Ni is reversible at lower amounts of sulfur. Nevertheless, at higher amounts of sulfur, bulk sulfidation happens that permanently damages the catalyst. Since the sensitivity of nickel to S poisoning lowers with increasing temperature, SOFCs can endure higher concentrations of S in the fuel feed compared to low-temperature fuel cells. Generally, the S amount in the fuel has to be brought down to ≤0.2 ppm or lower. There was an onset in performance degradation of Ni–YSZ when it was operated at

750, 900, and 1000 °C with fuel containing 0.05, 0.5, and 2 ppm of $H_2S$, respectively [241]. However, at a high concentration of $H_2S$ (>100 ppm), most of the SOFC anodes including Ni-doped ceria and sulfides were prone to poisoning in long run due to the formation of bulk Ni–S species [241–245].

Degradation was supposed to be generated by Ni surface reconstruction or S–electrolyte interactions. It is accepted that, due to higher sulfur surface coverage at higher temperatures, the density of Ni increased on the surface [246]. This will degrade the activity as steps are more active than terraces for $CH_4$ activation [244,245]. Additionally, YSZ promoted deactivation as it is more vulnerable to $H_2S$ compared to scandia-stabilized zirconia [247]. The ionic conductivity of the oxide phase plays a critical role in controlling sulfidation. As most of the anode materials including Pt are susceptible to sulfidation, developing S- and C-tolerant anode materials remains a challenge. To address this issue, researchers have explored several anode materials, and the details are presented below.

### 6.1. Oxide Anodes

In general, vanadates showed better sulfur tolerance among oxide anodes. Aguilar et al. [248–250] developed $La_xSr_{1-x}VO_{3-\delta}$ (LSV) for $H_2S$ fuel; it exhibited higher selectivity for $H_2S$ oxidation than $H_2$, and a maximum of 136 mWcm$^{-2}$ was achieved at 1000 °C with SOFC fabricated from LSV anode. This may be attributed to the improved adsorption process at the anode surface due to the fast response time with the introduction of $H_2S$ in the fuel stream. Further, a SOFC fabricated from $La_{0.7}Sr_{0.3}VO_3$ anode has shown better power density at the 10% $H_2S$ level. This is a $H_2S$ tolerance level 5000 times greater than the state-of-the-art Ni–based systems [249]. Similarly, a $Ce_{0.9}Sr_{0.1}Cr_{0.5}V_{0.5}O_3$ anode catalyst was used in $H_2S$–$CH_4$ and $H_2S$–$N_2$ fuel conditions. It was found that performance was about 85 mWcm$^{-2}$ for $H_2S$–$N_2$ fuel and was slightly lower for $H_2S$–$CH_4$ fuel [251]. To use $CH_4$ directly as fuel, Cu and Cu–Pd impregnated $La_{0.75}Sr_{0.25}Cr_{0.5}Mn_{0.5}O_{3-\delta}$ (LSCM) was used as SOFC anode [187]. Though the cell performed impressively in dry methane, the LSCM anode was not stable when exposed to $H_2$ with 50 ppm $H_2S$, and power density declined abruptly within 2 h. On the other hand, $La_{0.75}Sr_{0.25}Cr_{0.5}Fe_{0.5}O_{3-\delta}$ had good selectivity for $H_2S$. There was about three-fold increase in performance with $H_2S$ fuel [79]. Additionally, Ru-doped $Sr_{0.88}Y_{0.8}TiO_3$ showed good tolerance for sulfidation in 10–40 ppm $H_2S$ [190]. Mukundan et al. [252] explored a large number of perovskites and, among them, 50 wt% $Sr_{0.6}La_{0.4}TiO_3$–50 wt% YSZ anode was stable even when the hydrogen fuel contained 5000 ppm of $H_2S$. A similar anode was prepared by Cheng [253] using the solid-state method, and the cell in YSZ–LST/YSZ/Pt configuration was tested in 10,000 ppm $H_2S$-$H_2$ at 850 °C for >10 h, and it exhibited a power density of 132 mW cm$^{-2}$.

### 6.2. Cermet Anodes

Ni-based cermets are prone to sulfidation at high sulfur concentrations. Shatynski [254] studied transition metal sulfides such as $Ni_3S_2$ and $NiS_2$ possessing melting points of 1336 and 1554 °C, respectively, while $Ni_3S_4$ decomposed at 902 °C. The performance of conventional anode reduces rapidly in the presence of liquid $Ni_3S_2$. In 10 ppm $H_2S$ at 1346 °C, Ni reacts with sulfur, and liquid sulfide formation will decline. A Ni–GDC anode containing SOFC exhibited 10–12.5% degradation in $H_2$ fuel/200–240 ppm $H_2S$ over 500 h at 850 °C [243], and the degradation was ~20.6% for the conventional anode in $H_2$ fuel containing 50 ppm $H_2S$ in 120 h at 800 °C [255]. Lussier et al. [256] reported complete performance recovery for Ni–YSZ with shorter exposure times of $H_2S$, whereas it led to permanent degradation after ten hours of exposure. Zhang et al. [257] observed a decrease in the anode potential of the conventional anode electrodes from 0.61 to 0.34 V in presence of $H_2$ fuel consisting of 5 to 700 ppm $H_2S$. Nevertheless, the anode potential of Ni–GDC anode measured in pure $H_2$ decreased from 0.78 to 0.72 V under similar test conditions. The performance degradation was significantly lower in N–GDC anodes compared to the conventional anode in $H_2S$-containing fuels.

Brightman et al. [258] reported Ni–GDC degradation in the presence of 0.5 ppm $H_2S$, which was fully recovered upon the removal of $H_2S$. It was suggested that this initial poisoning behavior was due to adsorbed sulfur that inhibited the surface diffusion of H atoms to active sites. When Ni–GDC was exposed to 1–3 ppm of $H_2S$, secondary degradation was noticed due to increased ohmic resistance and was more acute at higher temperatures. The degradation could be reverted upon removal of sulfur. It was opined that the ohmic resistance increases as a result of surface microstructural changes in the Ni and/or CGO component of the cermet due to the dissolution of S at the surface.

Li et al. [259] evaluated the electrochemical performance of $BaCe_{0.9}Yb_{0.1}O_{3-\delta}$ (BCYb) nanoparticles infiltrated Ni–GDC anode. The maximum performances of the BCYb–Ni–GDC cell were 1.75 and 1.66 $Wcm^{-2}$ in pure $H_2$ and 500 ppm $H_2S/H_2$, respectively, at 650 °C. They also studied the effect of impregnation of $BaCe_{0.9}Yb_{0.1}O_{3-\delta}$ (BCYb) into the Ni–GDC cermet anode on sulfur poisoning resistance of Ni–GDC (NG) and Ni–GDC(NG) + BCYb anode-supported SOFCs. The NG and B + NG cells suffered a 51% and 5% decrease in power output after exposure to 500 ppm $H_2S/H_2$, respectively. This was attributed to the presence of BCYb nanoparticles and the high Fermi basicity and low electronic work function of proton-conducting doped barium cerate perovskites, which resulted in a strong tendency to adsorb and split water.

Lohsoontorn et al. [260] reported identical electrochemical behavior in Ni–CGO cermet anodes. Xu et al. [261] proposed the addition of the $Ni–CeO_2$ layer over the conventional anode, and the cells exhibited a modest increase of $H_2S$ tolerance but not as anticipated. The cell performance remained stable in the presence of $H_2S$ for about 2 h, and thereafter, it fell to 0 V in the subsequent 2 h. Additionally, the ScSZ electrolyte showed improved tolerance to $H_2S$ and the cell with Ni–ScSZ could tolerate up to 1 ppm $H_2S$ [262].

Grgicak et al. [263] analogized the performances of Ni–YSZ and Co–YSZ anodes in sulfur-containing fuel for a longer duration. Studies revealed stable performances of anodes over long periods in presence of $CH_4$ and $H_2$ fuels consisting of $H_2S$ due to the formation of NiS–YSZ and CoS–YSZ. Despite the early fall in SOFC performance when $H_2S$ was added to the fuel stream, the performance regained and surpassed the initial values after 3 h. The excellent performance of metal–sulfides proves beyond doubt that they are propitious anode materials for C- and S-tolerant SOFC systems. CoS–YSZ/YSZ/LSM operated with $CH_4/H_2S$ and air displayed notably larger exchange current densities when compared to the same system operated with hydrogen. Co-based anodes did not degrade when operated in $H_2S/CH_4$ for up to 6 days (Figure 16).

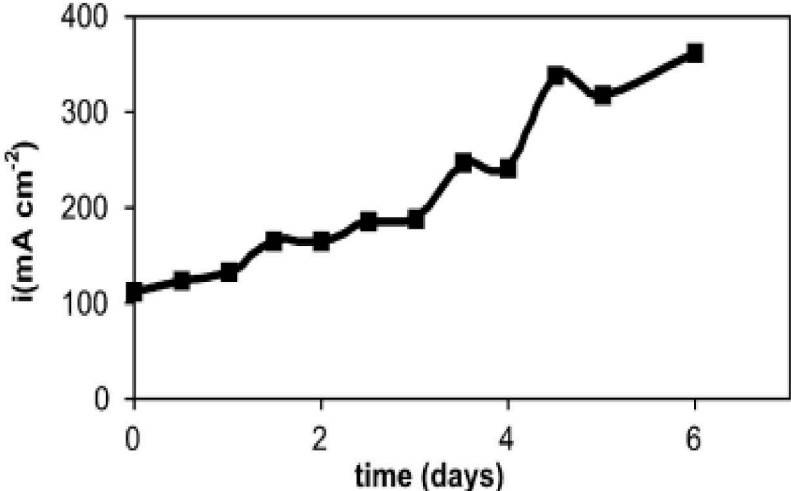

**Figure 16.** The plot of current density vs. time of a SOFC with Co-YSZ anode with 10% (*v/v*) $H_2S/CH_4$ at 0.5 V and 850 °C (adapted from [263]).

The Cu-based anodes are more tolerant to sulfur compared to Ni–YSZ cermet [264]. Generally, NiS is more stable than copper sulfides [265]. The surface sulfides are not likely to influence the performance of SOFC as Cu does not act as a catalyst in the anode. Similarly, Gorte et al. [183] operated a cell fabricated with a Cu–YSZ anode in a mixture of n–decane and 5000 ppm of sulfur mixed in the form of thiophene for 24 h without deactivation.

### 6.3. Bimetallic Cermets

Anode catalysts consisting of composite metal sulfides of Mo and other transition metals such as Fe, Co, and Ni were stable and promising for conversion of $H_2S$ in SOFCs [266]. Among them, Co–Mo–S along with 10% Ag showed improved activity and endurance [266]. Hua et al. [227] reported highly active Ni−Mo bimetallic alloys possessing thermal stability and sulfur/coke resistance for HC oxidation, and the SOFC exhibited a maximum power density of 594 mWcm$^{-2}$ at 800 °C in the presence of $CH_4$ fuel containing 50 ppm of $H_2S$. Furthermore, the Ni–Mo catalyst was stable, and the cell exhibited sustained and steady power output for longer durations.

Mukherjee et al. [267] observed efficient oxidation of $H_2$ at SOFC operating conditions Co-Ni anodes when compared to Cu anodes. The investigation of electrochemical oxidation of hydrogen with multifarious metal catalysts exhibited a volcano-type plot with Co and Ni occupying the place on either side of the maximum activity, indicating a probable symbiotic effect among them. Impressively, $Ni_{1-x}Co_x$–YSZ cermet SOFC anodes showed synergistic behavior for electrochemical oxidation of fuels containing $H_2S$; the performance exceeded those of individual sulfided Ni and Co anodes, thereby indicating a synergistic behavior in the Ni–Co–S anode [224]. The effect of adding Cu or Co to Ni–YSZ anodes on the SOFC performance depends on the catalytic activity of each metal. The sulfided $Ni_{0.69}Co_{0.31}$–YSZ exhibited improved activity in 10 vol% $H_2S/CH_4$. When a SOFC containing $Ni_{(1-x)}Cu_x$–YSZ anode was operated in $H_2S/CH_4$ atmosphere, the electrolyte cracked immediately due to mechanical stresses [224]. It was suggested that the anode and electrolyte TEC mismatch was more during the sulfidation of $Ni_{(1-x)}Cu_x$. However, it can be effectively addressed by introducing a graded functional layer.

Density functional theory calculations showed hollow sites of Ni(111) and Cu(111) surfaces were energetically favored by C and S. By adding Cu into Ni, a Ni–Cu alloy was formed, which reduced the adsorption of C and S possessing low adsorption energies because of reduced overlapping between the C 2p or S 3p and the metallic 3d orbitals [268]. Cu alloying with Ni reduces the adsorption energy and consequently the propensity of C or S adsorption on the Ni–Cu alloy surface. This clearly shows that Ni–Cu is a superior anode catalyst compared to the Ni anode with respect to carburization resistance and sulfur compliance.

Thus, anode cermets are found to be more promising for HC fuel. The use of a multi-component anode system becomes inevitable to meet all the required properties of HC-compatible SOFC. Moreover, anode cermet consists of metallic and ceramic phases, which need to perform synergistically. The homogeneity of anode composition is extremely important to meet such criteria. Further, anode microstructural properties such as surface area, particle size, and distribution need to be controlled to maximize the active TPB. Considering these facts, it is essential to adopt suitable synthesis techniques to produce anode cermet with improved properties.

## 7. Area Specific Resistance (ASR) of SOFCs Based on Hydrocarbon Compatible Anodes

The ASR is an important performance parameter besides the power density, especially in SOFCs, where the ohmic losses often dominate the overall polarization of the cell. The ASR in most cases is not very sensitive to small variations in cell voltage and fuel utilization and is less dependent on test schedules. By determining the ASR at a few different temperatures, apparent activation energy, Ea, may be derived. The area specific resistance (ASR), which is dependent on the current or the current density, can be determined from the difference in voltage V(j) divided by the corresponding difference in current density [269].

Overall, the ASR can be divided into Rs (ohmic resistance) and Rp (electrode polarization resistance). For a SOFC stack, ASR is defined as:

$$ASR = \frac{EMF - U}{i}$$

where EMF is the electromotive force with the inlet fuel and air, and U is the cell voltage at the current density, i, at the design point [270].

The ASR and power densities of some of the SOFCs fabricated from hydrocarbon compatible anodes along with the cell configuration, operating temperature, and fuels are summarized in Table 3.

**Table 3.** The ASR and power densities of SOFCs fabricated from hydrocarbon compatible anodes.

| Anode | Configuration | ASR ($\Omega cm^2$) | Temp. (°C) | Fuel | Power Density mW cm$^{-2}$ | Ref. |
|---|---|---|---|---|---|---|
| Ni/Ag/GDC | YSZ/LSCF-GDC | 1.12 | 750 | Syngas | 33 | [271] |
| $La_{0.20}Sr_{0.25}Ca_{0.45}TiO_3$ decorated with CGO coating and nanoparticles of Ni, Pt, or Rh. | 6ScSZ/LSM-YSZ | 0.78 | 850 | Syngas + 8 ppm $H_2S$ | - | [272] |
| Ni/10Sc1CeSZ | 10Sc1CeSZ symmetrical cell | 24.3 | 800 | $CH_4$ | - | [273] |
| $La_{0.65}Ce_{0.1}Sr_{0.25}Cr_{0.5}Mn_{0.5}O_{3-\delta}$ | ESC-YSZ | 1.6 | 900 | $CH_4$ | - | [274] |
| Ni- $Ce_{0.9}Sr_{0.1}Cr_{0.5}V_{0.5}O_3$-YSZ | ESC-YSZ | 5.9 | 850 | $H_2S$-$CH_4$ | 25 | [251] |
| $La_{0.8}Sr_{0.2}Sc_xMn_{1-x}O_3$–GDC-Pd | Three-layer ESC infiltration | 0.4 | 700 | $CH_4$ | 341 | [139] |
| $La_{0.75}Sr_{0.25}Cr_{0.5}Mn_{0.5}O_{3-\delta}$ | ESC-YSZ-symmetrical cell | 0.613 | 950 | $CH_4$ | 347 | [74] |
| LSCM-0.5%Pd-5%$CeO_2$ | Three-layer ESC infiltration | 0.14 | 800 | $CH_4$ | 710 | [275] |
| $La_{0.75}Sr_{0.25}Cr_{0.5}Mn_{0.5}O_{3-\delta}$ | ESC | 2.3 | 800 | Ethanol | 101 | [72] |
| $(La_{0.75}Sr_{0.25})Cr_{0.5}Mn_{0.5}O_{3-\delta}$ | ESC-YSZ-Symmetric | 0.3 | 900 | $CH_4$ | 230 | [73] |
| $La_{0.8}Sr_{0.2}Sc_xMn_{1-x}O_3$–GDC-Pd | Three-layer ES infiltration | 0.4 | 700 | $CH_4$ | 341 | [163] |
| $Sr_{0.6}La_{0.4}TiO_3$-50% $CeO_2$ | ESYSZ | 2 | 900 | $CH_4$ | 139.6 | [276] |
| $La_{0.2}Sr_{0.7}TiO_3$-GDC-Cu | Three-layer ESC infiltration | 0.15 | 750 | $CH_4$ | 540 | [100] |
| 10%Ni-LSFC | ES-CGO | 0.33 | 800 | Propane | 421 | [277] |
| $(Pr_{0.75}Sr_{0.25})_{1-x}Cr_{0.5}Mn_{0.5}O_{3-\delta}$ | ES-YSZ | 3.52 | 910 | $CH_4$ | 18 | [278] |
| 1:1 Co-Cu-$CeO_2$ | ESC-YSZ | 3.72 | 800 | $CH_4$ | 67 | [279] |
| 50:50 CoCu-$CeO_2$-YSZ | 3-layer-YSZ | 0.8 | 800 | $CH_4$ | 250 | [221] |
| $Cu_{1.3}Mn_{1.7}O_4$ on Ni–SDC | ASC-SDC | 0.120 | 700 | $CH_4$ | 375 | [280] |

It is worth noting that the ASR of ASC is lesser than ESC in the intermediate temperature regime. The anodes with Nobel metals such as Pt, Pd, etc., yielded lower ASR. As such, it is difficult to draw a direct comparison based on ASR reports, as different researchers have reported different forms of ASR such as ASR cell (includes electrolyte ohmic + cathode ASR + anode ASR) [279], polarization ASR (non-ohmic anode + non-ohmic cathode) [278], anode ASR (includes ohmic anode + non-ohmic anode), etc. One possible way to study anode ASR is by performing symmetric cell analysis. However, such studies report extraordinarily high anode ASR [273] as kinetics in symmetric cells may not depict the actual cell conditions (non-ohmic anode ASR).

## 8. Synthesis of Anode Composites

Several methods have been adopted for the synthesis of SOFC anodes as shown in the flowchart (Figure 17) and Figure 18 shows the schematic of the most widely used methods for the fabrication of the anodes [281]. The methods may be classified as (i) powder route-based, (ii) coating-based, (iii) infiltration–based, and (iv) mechano-fusion-based. Studies have shown that, by controlling the particle size and particle size distribution, optimum properties for the SOFC electrode application can be achieved. More details have been discussed based on the above classification.

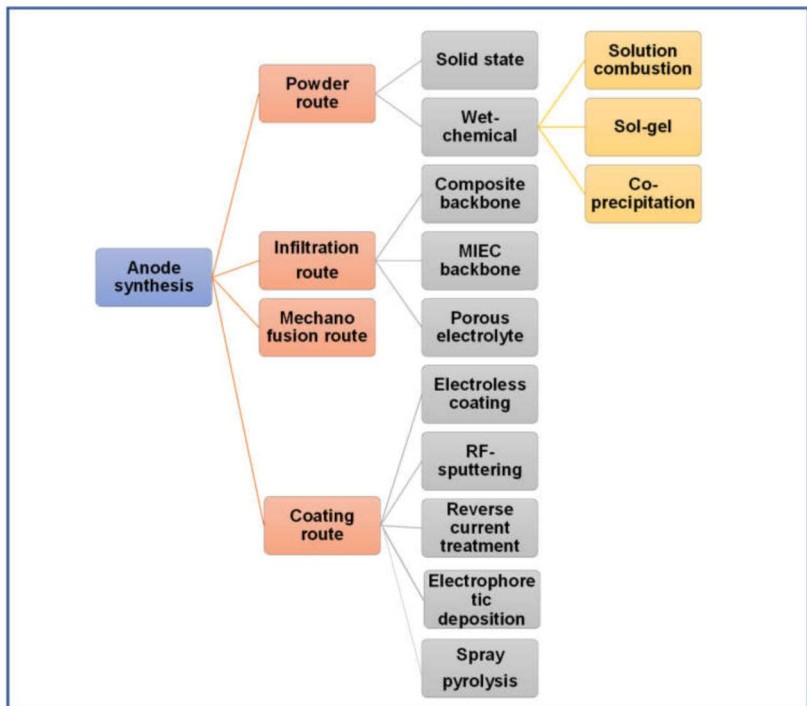

**Figure 17.** Flowchart showing various routes used for anode synthesis.

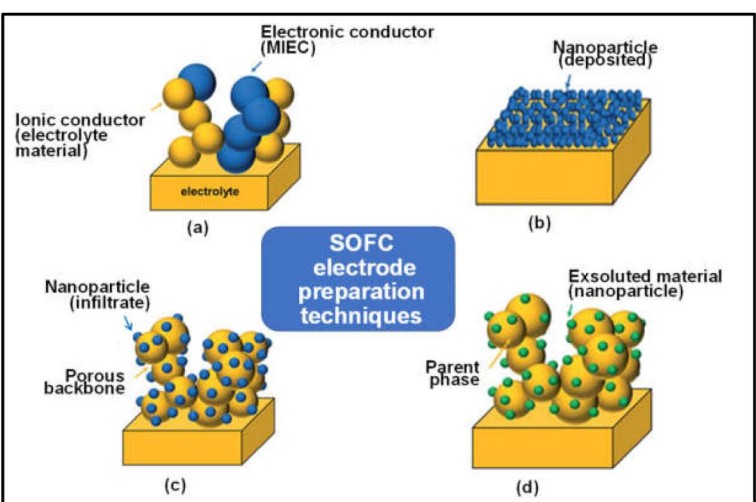

**Figure 18.** Schematic of various methods adopted for the fabrication of composite electrode using (**a**) the conventional ceramic processing method, (**b**) thin-film deposition, (**c**) infiltration, and (**d**) ex-solution (adapted from [281]).

### 8.1. Powder Route

8.1.1. Solid-State Method

The most commonly used conventional method for the synthesis of the anode is the solid-state reaction route [225]. It involves the physical mixing of a mixture containing oxide powder in a desirable stoichiometric ratio. Mixing is carried out in a ball milling machine with a suitable solvent. Even in the case of cermet, only metal oxides are used in combination with ionic conducting oxide phase during the synthesis which is reduced in $H_2$ atmosphere to form the cermet at a later stage. Thus, prepared oxide compositions are calcined and then utilized in the single-cell SOFC fabrication process. However, the major disadvantage of this process is the difficulty in controlling the particle size and its distribution. Always, a smaller particle size is advantageous as it can extend the TPB length, which, in turn, increases the electrochemical active zone [226]. Similarly, good distribution improves the homogenization and, hence, the efficiency of the catalyst can be improved. Particularly for the multi-component anode system, homogeneity is considered the most important criterion.

8.1.2. Wet Chemical Routes

The wet chemical routes are promising as they involve atomic-level mixing and yield more homogeneous powders compared to those prepared by the solid-state method. The most commonly used wet chemical techniques for the synthesis of nanocomposites are

- Solution combustion synthesis
- Sol–gel synthesis and
- Co-precipitation synthesis
- Hydrothermal method.

Each method has its advantages and limitations. Figure 19 shows the advantages and disadvantages of the prominent wet-chemical routes.

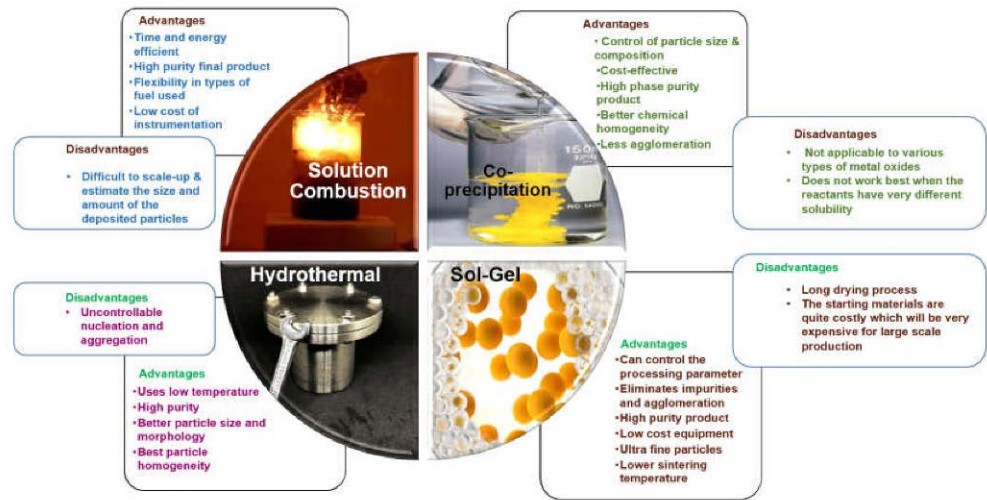

**Figure 19.** Schematic showing the advantages and disadvantages of the most widely used wet-chemical routes.

Prasad et al. [282] reported a single-step glycine nitrate-based solution combustion method for the synthesis of NiO–CGO composite powder. TEM analysis of the powder revealed the growth of nickel nanoparticles from 5–10 to 20–40 nm under steam-rich conditions because of the sintering of nickel nanoparticles that was responsible for decreased reforming activity. However, under steam lean conditions, nickel nanoparticles size was confined to <10 nm and thus exhibited a steady and medium reforming activity. This clearly indicates steam as the main reason for the sintering of nickel nanoparticles. Additionally, TEM/EDS analysis on the spent catalysts demonstrated the location of nickel

nanoparticles on the CGO support surface revealing close contact with the support. However, the surface morphology of the Ni-CGO cermet was not modified drastically after reduction as well as after the catalytic tests (Figure 20). This can subdue carburization by sustaining good metal (Ni)–support (CGO) interaction even under steam lean conditions. Similarly, Osinkin et al. [283] prepared high-performance anode-supported SOFC with Ni–YSZ–$Zr_{0.83}Sc_{0.16}Ce_{0.01}O_{1.92}$ anode synthesized by glycine nitrate process. Further, these anodes were impregnated with cerium and praseodymium oxides to improve electrode catalysis. Thus, the fabricated cell yielded 1.25 $Wcm^{-2}$ at 700 °C and 2.5 $Wcm^{-2}$ at 900 °C in $H_2$.

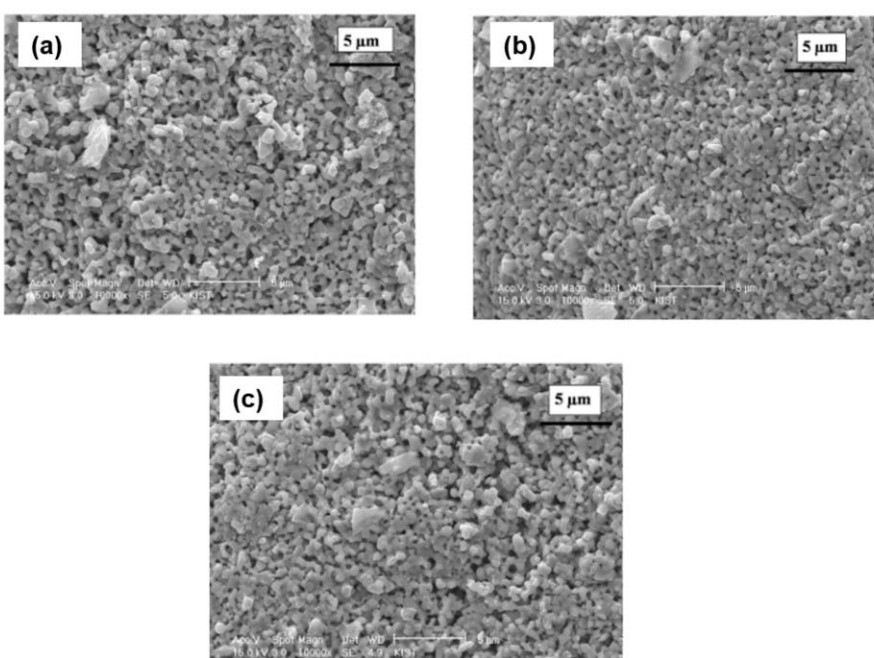

**Figure 20.** Scanning electron micrographs of Ni-CGO cermet catalysts (**a**) after reduction and after catalytic tests at (**b**) S/C = 0.5 and (**c**) S/C = 1.5 at 800 °C for 5 h (adapted from [282]).

Razpotnik et al. [284] synthesized NiO–YSZ using a modified Pechini gel route by using ascorbic acid as the complexation agent. Modifying the Pechini-type sol–gel method has been employed for synthesizing NiO–YSZ powders with varying particle sizes (nm to μm). The fuels used in the reaction influenced the surface area of the synthesized anode. Ascorbic acid was advantageous in synthesizing a higher surface area anode powder compared to the powder synthesized using citric acid. The metal ion concentration influenced the crystallite size of the products, and there was a reduction in the grain size when the metal concentration was lower. Lower metal concentration yielded anode powders possessing 200 to 470 nm crystallite size and 20 $m^2g^{-1}$ surface area, which, upon pelletization, could be sintered at a low temperature (1300 °C).

Cela et al. [285] synthesized the NiO–CGO ($Ce_{0.9}Gd_{0.1}O_{1.95}$) composite by single and double steps modified sol–gel route. In the first step, precursor resins of CGO and NiO phases were synthesized separately by the polymeric precursor method and were mixed to obtain a single homogeneous resin. In contrast, the composite obtained by the two-step synthesis procedure involved the synthesis of two separate polymeric precursor syntheses of NiO and CGO. In the next step, CGO and NiO powders were mixed in a 1:1 weight ratio followed by ball milling. The cell voltages and power densities of the single cells prepared from NiO–CGO using the single-step method exhibited maximum power densities compared to the cells synthesized from the two-step synthesized anode and conventional anode. This is attributed to the homogeneous dispersion of the nickel phase and homogeneous porous microstructure that improved the performance in the SOFC fabricated from one-step synthesized anode.

Suciu et al. [286] discussed the single-step sol–gel method for the synthesis of SOFC anodes. Two different compositions of binary mixtures of NiO and YSZ particles were prepared by the sol–gel method using sucrose and pectin as organic precursors. It facilitated the synthesis of the composite of NiO and YSZ with a uniform particle size of 30 nm at a lower calcination temperature of 800 °C and in an economical way. Additionally, the particles had quite uniform particle sizes within each type.

The easily scalable co-precipitation method is being employed for the synthesis of ultrafine powders with uniform composition. Studies have shown that the composition and microstructure of the powders are influenced mainly by the precipitating agent [287–289]. Studies have shown that precipitation agents greatly influence the composition and microstructure of NiO–YSZ powders synthesized by co-precipitation method [287–289]. The urea hydrolysis method also yields precipitates that can be easily controlled by the concentration of urea and time of hydrolysis. Lin et al. [290] utilized the urea hydrolysis technique to synthesize the NiO–YSZ nanocomposite. The results showed rather a uniform microstructure of Ni/YSZ cermets containing uniformly dispersed submicron-sized Ni particles surrounded by fine pores within the YSZ matrix.

Lee et al. [291] hypothesized that the high activity for CO oxidation of gadolinia-doped ceria (GDC)-coated Ni catalysts can obstruct carbon formations due to CO disproportionation at lower temperatures. GDC-coated Ni catalyst prepared using the hydrothermal route was used as a catalyst layer in GDC-based SOFCs. The SOFC displayed a power density of 1.42 $Wcm^{-2}$ at 610 °C in $CH_4$ devoid of carbon development. In particular, the GDC-based low-temperature SOFCs showed durability for 1000 h at a high current density of 1.2 $A/cm^2$. Further by utilizing catalytic partial oxidation of fuel, carbon coking was eliminated on Ni and facilitated the oxidation of methane [292]. Ni–GDC anode-based SOFCs showed excellent power densities of 1.35 and 0.74 $Wcm^{-2}$, respectively, at 650 and 550 °C for over 500 h.

Kumar et al. [293] have synthesized Cu-doped Ni–YSZ–GDC multicomponent cermet powder with the composition $Ni_{0.9}$–$Cu_{0.1}$–$YSZ_{0.95}$–$GDC_{0.05}$ in a single pot using the versatile solution combustion method employing citric acid and hexamine as fuels. The study revealed that citric acid was a better fuel for obtaining agglomerate-free anode powder, and carbon deposition was reduced by 62% with this anode. The synthesized powder was tapecast, and ESC and ASC were fabricated using the fabricated anode tape. By using the SCS anode as a functional layer in ASC, a maximum power density of 884 $mWcm^{-2}$ was reported at 800 °C using $CH_4$ as the fuel. Interestingly, the SCS anode displayed similar exchange current density and anode impedance along with excellent electrocatalytic activity to $CH_4$ and hence can be employed as a propitious anode functional layer.

### 8.2. In Situ Exsolution Route

Surfaces embellished with homogeneously dispersed catalytically active nanoparticles are instrumental in areas such as catalysis and renewable energy, and in situ exsolution is deliberated as an efficient synthesis route for nanostructured composite electrode materials for the development of next-generation electrochemical devices for energy conversion [294]. In this method, (electro)catalytic elements are formed in the crystal lattice during the synthesis step under oxidizing conditions that forms a solid solution and precipitate (exsolve) on the surface of the oxide phase on heating the sample in a reducing atmosphere at $T \geq 800$ °C [295]. The formed nanoparticles are more uniformly dispersed and possess better thermal stability than those formed by the conventional impregnation method. This technique has gained attention in recent years due to its ability for in situ generation and regeneration of nanoparticles to improve anode catalysis. More particularly, this technique is useful in controlling the metal agglomeration of the nano-anode composite. In this regard, perovskites containing easily reducible cations were used in the anode

composite, which, in turn, reduced in the high temperature reducing environment, as shown in Equation (18) [295].

$$\text{Reduction}: \ M_xO_y + \frac{y}{2}H_2 \ \rightarrow \ xM^o + yH_2O \tag{18}$$

SOFCs fabricated with the stoichiometric $(La_{0.7}Sr_{0.3})CrO_3$ (LSC)-Ni anode and the A-site-deficient LSCNi anode subjected to in situ exsolution of nano-Ni, respectively, displayed maximum power density of 135 and 460 $mWcm^{-2}$ in 5000 ppm $H_2S$–$H_2$ [295]. Additionally, the SOFC also demonstrated favorable redox stability in fuel with a significant amount of $H_2S$. The introduction of A-site deficiency can help in the formation of high mobility oxygen vacancies and prevents Ni nanoparticles from oxidizing, thus considerably increasing the electronic conductivity and catalytic activity simultaneously.

Madsen et al. [296] prepared lanthanum chromite $(La_{0.8}Sr_{0.2}Cr_{1-y}X_yO_{3-\delta}$, X = Ni, Ru) and the transmission electron microscopy and X-ray photoelectron spectroscopy studies showed the formation of Ni or Ru metal nano-clusters on oxide surfaces during the initial stages of SOFC after exposure to hydrogen at 800 °C. The precipitated Ru and Ni nano-clusters enhanced the cell performance and decreased anode polarization resistance during the first 50–100 h of SOFC operation. The possibility of regeneration of the catalyst by oxidizing the reduced material, causing the catalyst metal to re-dissolve into the oxide and then reducing again and causing precipitation of fresh metal nano-clusters, was suggested. Equivalent regeneration may be possible in SOFC anodes.

By doping Ni using an incipient wetness method in lanthanum ferrites such as $La_{0.6}Sr_{0.4}Fe_{0.8}Co_{0.2}O_3$, followed by heat treatment, exsolved perovskites can be synthesized [297]. The heat treatments that assist the exsolution process involve calcination at 500 °C in the air followed by a reduction in diluted $H_2$ at 800 °C. These processes permit the formation of a dual-phase material consisting of a Ruddlesden–Popper-type structure and a solid oxide solution such as $\alpha$-Fe100-y-zCoyNizOx oxide.

The electrochemical performance of SOFC and catalytic activity under $CH_4$ operation was enhanced by doping Ce in nickel-doped $La_{0.7}Sr_{0.3}FeO_{3-\delta}$ (LSFNi) to obtain $La_{0.6}Ce_{0.1}Sr_{0.3}Fe_{0.9}Ni_{0.1}O_{3-\delta}$, (CLSFNi) [298]. Under reducing conditions, the electrode material was converted into a $LaFeO_3$ perovskite main phase with a minor amount of $SrLaFeO_4$ phase along with a Ni–Fe alloy secondary phase. During this process, many nanoparticles are exsolved from the electrode surface, and they can significantly lower the polarization resistance of the anode and increase the cell performance. Symmetric cells fabricated from LSFNi and CLSFNi exhibited very high-power density (900 $mWcm^{-2}$ at 850 °C). By doping Ce in CLSFNi, the methane reforming activity improved and greatly improved the performance of the CLSFNi electrode (522 $mWcm^{-2}$) over the LSFNi electrode (221 $mWcm^{-2}$) in wet $CH_4$/air (3% $H_2O$) at 850 °C. However, the cell with in situ exsolved $Ni-Ba-(Ce_{0.9}Y_{0.1})_{0.8}Ni_{0.2}O_{3-\delta} + Gd_{0.1}Ce_{0.9}O_{1.9}$ (Ni−BCYN+GDC) perovskite anode yielded a moderate power output of 80 $mWcm^{-2}$ at 750 °C in wet $CH_4$ (with 3% $H_2O$) [299].

Wan et al. [127] utilized a 10% surplus Ni-doped $SrV_{0.5}Mo_{0.5}O_{4-\delta}$ scheelite structure oxide as the anode. During reduction, $SrV_{0.5}Mo_{0.5}O_{4-\delta}$ transformed into a cubic perovskite structure $(SrV_{0.5}Mo_{0.5}O_{3-\delta})$ with a space group of $Pm\bar{3}m$ and a minor Ni, with Ni exsolution from the oxide lattice. The exsolved nickel nanoparticles enhanced the catalytic activity for electrochemical oxidation reaction to a greater extent and produced 0.56 $Wcm^{-2}$ at 800 °C. Similarly, FeRu alloy (FRA) nanoparticles' surface with Ruddlesden–Popper (RP) type layer perovskite $PrSrFe_{1-x}Ru_xO_{4+\delta}$ (RP–PSFeRu) was synthesized by an in situ reduction of the cubic $(Pr_{0.5}Sr_{0.5})_{0.9}$-$Fe_{0.9}Ru_{0.1}O_{3-\delta}$ (PePSFeRu) in $H_2$ at 800 °C (Figure 21) [300]. The $La_{0.8}Sr_{0.2}Ga_{0.83}Mg_{0.17}O_{3-\delta}$ (LSGM) ESC with RP–PSFeRu–FRA–GDC composite anode delivered maximum power densities of 0.75 and 0.50 $Wcm^{-2}$ in wet $H_2$ and $C_3H_8$ at 800 °C, respectively, and it exhibited exemplary stability in a wet $C_3H_8$ atmosphere. Additionally, CoFe nanoalloy catalysts embedded $Sr_3FeMoO_7$ oxide was explored as the SOFC anode [301]. The composite anode consisting of 50 wt% CoFe-$Sr_3FeMoO_7$-50 wt%

$Sm_{0.2}Ce_{0.8}O_{1.9}$. showed excellent performance ($1 Wcm^{-2}$ at 850 °C) and carbon resistance in 40% $C_3H_8$ fuel [301].

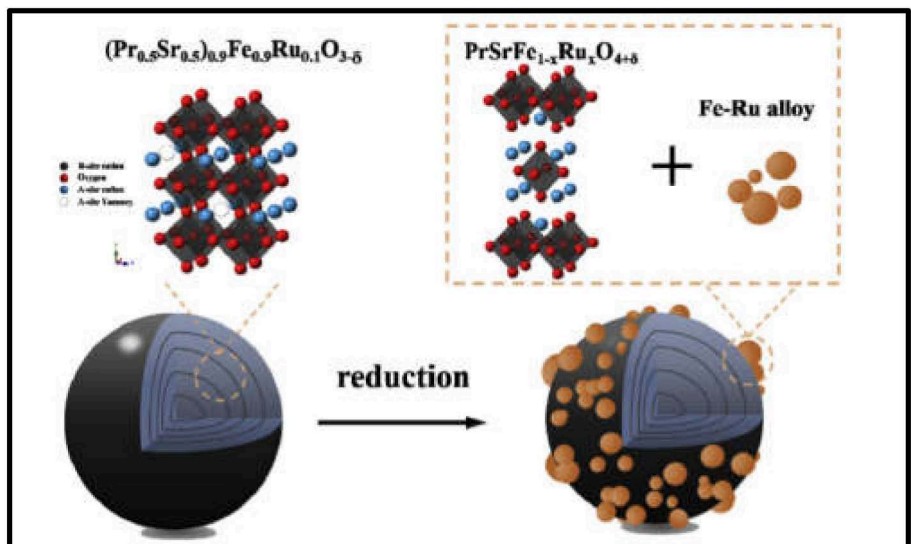

**Figure 21.** Schematic diagram of the morphology and crystal structure evolution of PePSFeRu after reduction (adapted from [300]).

Liu et al. [302] attempted a $Ba(Ce_{0.9}Y_{0.1})_{0.8}Ni_{0.2}O_{3-\delta}$ (BCYN) decorated $Gd_{0.1}Ce_{0.9}O_{1.95}$ (GDC) composite anode by solution impregnation. The metal Ni nanoparticles are in situ exsolved from BCYN catalyst during reduction. The ESC with Ni–BCYN/GDC composite anode yielded peak power density up to 211 $mWcm^{-2}$ in $CH_4$ at 750 °C. Thus, in situ exsolution is being recognized as a more efficient approach to preparing nanoparticle-decorated ceramic SOFC anodes.

Qin et al. [303] developed Ru/Nb co-doped $(Pr_{0.5}Sr_{0.5})_{0.9}Fe_{0.8}Ru_{0.1}Nb_{0.1}O_{3-\delta}$ (PSFRN) cubic perovskite oxide, which, after treating in wet $H_2$ at 900 °C for 2 h, changed into RP layered perovskite $PrSrFe_{0.8}Ru_{0.1}Nb_{0.1}O_{4+\delta}$ (RP-PSFRN). $Fe_{0.7}Ru_{0.3}$ alloy–FeOx oxide core-shell nanoparticles were in situ exsolved on the reduced-PSFRN (R-PSFRN) (Figure 22). The SOFC exhibited a peak power density of 0.683 and 0.537 W $cm^{-2}$ in wet $H_2$ and $C_3H_8$ as fuels at 800 °C, respectively. It also exhibited a stable output under a constant current load of 0.15 A $cm^{-2}$ in $C_3H_8$, illustrating a high resistance to carbon deposition and coking.

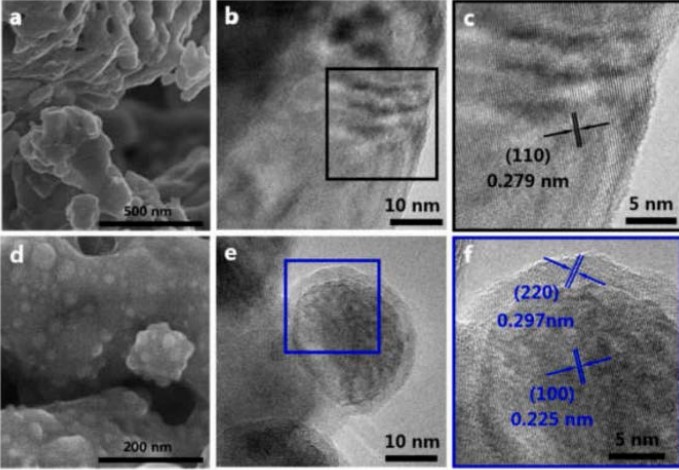

**Figure 22.** (**a**) SEM and (**b**,**c**) TEM images of as-prepared PSFRN. (**d**) SEM and (**e**,**f**) TEM images of R-PSFRN (adapted from ref. [303]).

*8.3. Infiltration*

The infiltration technique is one of the extensively prescribed methods for the fabrication of nanostructured anode composite. Infiltration of metal salt solution in the later stage of the fabrication process can eliminate the risk of metal sintering and circumvent solid-state reaction among the electrolyte and electrode materials that might otherwise happen at higher processing temperatures (1100–1300 °C). The infiltration method is further classified into three types based on backbone structure.

8.3.1. Infiltrated Porous Electrolyte Backbone Electrodes

This type of backbone for electrodes is constituted by the backbone grains consisting of a similar electrolyte material. Nevertheless, since the electrolyte is not involved in electron conduction and is mainly creating ion-conducting pathways, it is a requisite to infiltrate a material that can create electron pathways along with the electrocatalytic sites within the electrode. The important edge is the capability to infiltrate better electrocatalysts and to avoid conventional high-temperature sintering. Further, the ceramic phase in composite anodes preferably provides a sturdy network that partially slows down the Ni agglomeration compared to the sintering of pure Ni particles. Therefore, in metal cermets, metal agglomeration relies on the continuous particle size distribution of the ceramic phase and the volume ratio between metal and ceramic [304–306]. Qiao et al. [307] impregnated Ni/Ni–CeO$_2$ into the porous YSZ matrix. The process has reduced the Ni content in the anode to 25 wt% with improved cell performance. Li et al. [308] have investigated the effect of the addition of fuel such as urea to the impregnated solution and found a 36% increase in the performance of SOFC with the addition of urea. In this case, urea was added with the impregnating precursor to tailor the distribution and morphology of Cu in the cermet anode. The precursor was infiltrated, and CuO was formed inside the porous YSZ matrix by in situ combustion synthesis. It resulted in the formation of a well-connected conducting network of Cu. This, in turn, improved the performance of the cell.

8.3.2. Infiltrated Mixed Ionic and Electronic Conductors (MIEC) Backbone Electrodes

The capability of MIECs to facilitate both electron and oxygen ion conduction channels via an electrode permits them to be used as single-component electrodes. Even these electrodes can be improved by infiltrating a secondary electrocatalyst into the MIEC backbone. These types of electrode architectures are much common in the cathode than in the anode.

8.3.3. Composite Backbone Electrodes

The embodiment of nanoparticulate networks into working composite electrodes will increase the electrocatalysis and can broaden the strict TPB reaction area. The infiltrated nanoparticulate network layer does not need to be continuous throughout the electrodes as they already have integrated electronic and ionic percolation networks throughout the electrodes, which reduces the reliance of the SOFC to a greater extent on the stability of the morphology of the nanoparticulate networks due to the existence of a structurally stable backbone. It would be the preferred architecture for an anode composite, as it can take care of metal agglomeration in the typical SOFC operating condition.

Sadykov et al. [309] developed nanocomposite catalysts comprising Ni particles incorporated into the complex oxide matrix of YSZ or ScSZ integrated with doped ceria–zirconia oxides or La–Pr–Mn–Cr–O perovskite and promoted by Pt, Pd, or Ru. Structured nanocomposite anodes prepared using the infiltration route showed high efficiency and stability in the reactions of steam reforming of methane as well as oxygenates (ethanol, acetone).

Zhan et al. [310] explored a new approach to produce thin lanthanum strontium gallium manganite (LSGM) electrolyte-based SOFC without a lanthanum-doped ceria (LDC) barrier, by employing economical high-temperature firing. The electrolyte was assisted by a porous Ni-impregnated LSGM substrate that curtailed unfavorable Ni–LSGM interaction. Another important attribute of impregnation is the formation of a nanoporous

metal layer on the porous LSGM substrate surfaces. The metallic nanoparticles (<100 nm) yielded a TPB density of $37.15/\mu m^2$ and resulted in a low anode polarization of $0.026 \ \Omega cm^2$ at 650 °C. The total resistance of this cell configuration was much less, creating this as a stimulating new route for ascertaining reduced temperature SOFCs.

Jiang [311] has highlighted the issues associated with wet impregnation. Although the wet impregnation method can be adopted into currently used SOFC fabrication steps, it adds extra processing and sintering steps. This will become a hurdle if higher oxide loading is required as evident from the following example: the wet impregnation process was performed six times to achieve a GDC vol% of 37 in LSM [312]. This will definitely increase the fabrication time and cost of the cell. The additional cost may be reduced by optimizing and incorporating automation in the fabrication. Further, whenever thick porous structures have to be impregnated, the capillary forces will not be sufficient for the impregnation. Park et al. [313] observed a drastic fall in the impregnated $CeO_2$ and Cu content in the anode support in areas near the electrode/electrolyte interface. However, during prolonged operation, there was a decline in their microstructure stability and performance. As the impregnated phase is very fine (particle size: 100–300 nm), it tends to sinter, resulting in higher grain growth due to the larger surface energy of the nanosized oxide or metallic phase. The anode composite containing nanosized Ni, Pt, and Pd showed poor thermal stability and underwent higher sintering and grain growth even at temperatures as low as 700–800 °C [314–316]. The pure Cu/ceria-based composite anode will be unstable in SOFC operation conditions as the structure depends on HC deposits to provide electronic connectivity. The cell performance of precious metal-impregnated ceria-based composite anodes operated for 100 h with dry $CH_4$ decreased by 15%, and impedance results showed enhancement in the cell ohmic resistance, suggesting loss of carbon [317]. To date, all the studies reported so far are based on button cell testing, and it has to be noted that the efficiency of the small cells will not scale up proportionately when the cell area is increased [318]. Therefore, to justify the wet impregnation technique as a relevant and compatible method for the development of SOFCs, the performance and stability of large cells and stack level testing have to be validated. Venâncio et al. [319] prepared the single cell using the wet impregnation method in a YSZ electrolyte-based scaffold. An aqueous metal nitrate solution corresponding to the composition of $La_{1.5}Al_{0.33}Mn_{0.17}O_{3-\delta}$ (LAMO) was infiltrated in 8YSZ. The cells containing infiltrated 50 wt% LAMO/YSZ anode delivered an output of $150 \ mWcm^{-2}$ using methane as fuel.

### 8.4. Mechanofusion

Mechanofusion is a dry process performed in a rotating reactor (>1000 rpm) consisting of a cylindrical chamber equipped inside with compression tools and blades [320,321] (Figure 23). The optimized anode compositions are introduced into the chamber, and the chamber is rotated; during this process, the particles are pressed together and to the chamber walls. Adhesion between the constituents of anode composite is achieved with the compression tools and the centrifugal force created by the high rotation speeds.

Fukui et al. [320] proposed the mechanofusion technique to control the morphology of Ni–YSZ anode composite. NiO and YSZ powder were mechanically processed by using high shear and compression force. It led to the formation of microporous Ni–YSZ cermet anode containing well-dispersed fine YSZ and Ni grains, which increased the TPB length and, in turn, led to the exemplary performance of the anode. Similarly, Misono et al. [322] developed Ni–GDC by the mechano-chemical route, and the electrode polarization was brought down to almost half to that of the composite synthesized by the normal solid-state route.

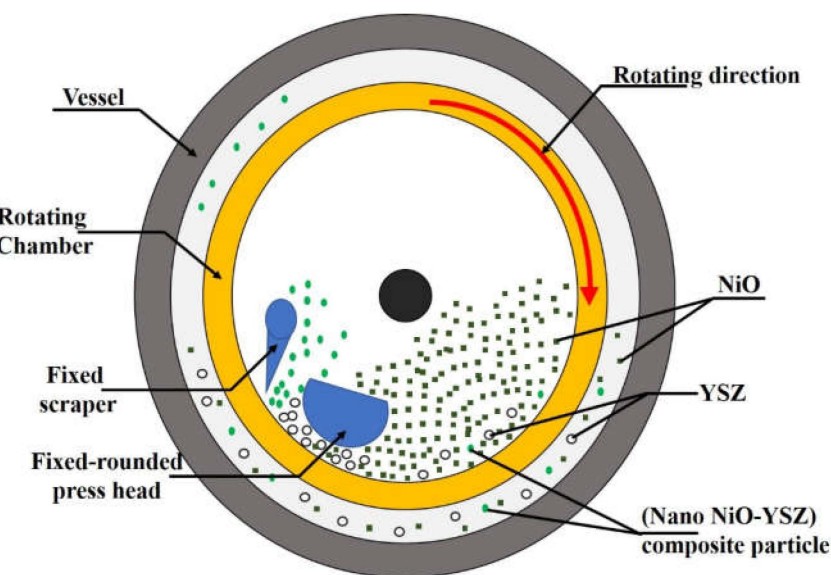

**Figure 23.** Schematic of the mechanofusion system (redrawn after [321]).

*8.5. Coating Routes*

8.5.1. Electroless Coating

Electroless deposition is more versatile since all metals of anode composite can be deposited. Any anode composition can be prepared as long as the overall redox reaction of the reducible metal salt by the reducing agent is thermodynamically favorable [323]. Additionally, electroless deposition yields metal nanoparticles and does not require further processing steps.

Rahman et al. [324] utilized the electroless coating technique to coat nanosized 8YSZ powders (<70 nm) with metallic Ni for the SOFC anode using sodium phosphinate mono-hydrate as the reducing agent without the aid of any sensitizing agent. The obtained powder contained a homogeneous mixture of nanosized Ni and YSZ particles. Thus, developed cermets showed conductivity percolation at room temperature as well as at elevated temperatures. However, this technique has not been employed for the synthesis of other metal cermet anodes.

8.5.2. RF-Sputtering

Sputtering is a type of physical vapor deposition process and is capable of producing dense and porous thin films depending upon the process parameters [325]. In the sputtering process, the material is dislodged by bombarding a solid cathode target with positively charged ions from a noble gas discharge source. The generated ions move with high kinetic energy and collide with the surface break free, or sputter, atoms of the cathode during the momentum transfer process. In DC sputtering, a high negative voltage (3–5 kV) is applied to the target placed in the vacuum chamber from an external source; while the chamber and the substrate are ground. Argon is passed between the target and substrate that is subjected to an electric field capable of inducing ionization and generating plasma. The sputtering process happens when the positive ions are accelerated towards the negatively charged target. In the case of metallic targets, a DC source is used for providing the potential at the cathode. For sputtering dielectric materials, a radio frequency (RF) source is used. To achieve higher sputtering efficiency, a magnetron source is employed [325].

Lao et al. [326] have developed a porous Ni–YSZ electrode along with dense YSZ electrolyte by RF sputtering. It is believed that these bilayered structures are endowed with the desired nano-sized grains in the anode that leads to increased TPBs, in turn, reducing the anodic overpotential. Additionally, these layers did not exhibit any delamination or cracking along the bilayer interface, which are required for decreased contact resistance in the SOFC.

Jou et al. [327] employed reactive magnetron co-sputtering of Ni and Zr–Y targets to produce thin, stable, and nano-porous Ni–YSZ films that displayed poor performance compared to screen-printed thick anode film probably due to the lower thickness. A single-cell SOFC containing a 0.7 μm thin anode and 31% porosity showed a power density of 0.38 mWcm$^{-2}$ vis-à-vis 0.76 Wcm$^{-2}$ power density of a cell containing a thick screen-printed anode (20 μm) with 36% porosity. A single-cell SOFC containing three anode layers with 3.1 μm thickness and 35% porosity displayed a power density in the range of 0.6 to 1.4 mWcm$^{-2}$ [315]. Thus, it is evident that such an anode may be applicable as an intermediate anode layer to enhance the performance of SOFCs. Similarly, Rezugina et al. [328] demonstrated that pulsed DC magnetron reactive sputtering to be an efficient technique for developing thin conductive films of Ni–YSZ with a deposition rate of 4 μmh$^{-1}$.

### 8.5.3. Reverse Current Treatment

Klotz et al. [329] proposed the reverse current treatment (RCT) technique, which resulted in the in situ formation of nanostructures at the interface of electrolyte/anode, of an ASC. A nano-structured interlayer of ~200 nm thickness was formed after multiple short-time reverse current treatment. In this regard, reverse current with a magnitude of 2 Acm$^{-2}$ was applied for 10 s at 700 °C. A detailed parameter variation revealed the following effects occurring during reverse current induced reduction and subsequent reoxidation of the electrolyte material YSZ.

- Identical RCT leads to the same performance improvement of ASC cells (Total ASR was reduced by ~10%) between 675 and 725 °C.
- Below 0.005 atm pH$_2$O, the reoxidation process takes place much slower than at 0.05 atm pH$_2$O and does not build up a high-performing nano-structured interlayer.
- Above 0.05 atm pH$_2$O, humidity prevents the ASC cell to reach the required decomposition voltage of the electrolyte material YSZ, and no change in anode performance is detectable.
- Single RCT with a duration of 40 s for the applied reverse current induces thickest and best-performing nano-structured interlayers.

The duration of the reverse current applied is identified as a crucial parameter, leading to the highest performance enhancement by RCT. A duration of 40 s resulted in the highest performance enhancement ever achieved through this method.

### 8.5.4. Electrophoretic Deposition

Electrophoretic deposition (EPD) is a common, simple, and economical colloidal processing technique that employs electrophoresis for the movement of suspended charged particles in an electric field. There are a large number of review articles related to electrophoretic deposition for SOFC [330–333]. About 250 papers have been published on electrophoretic deposition in SOFC, and most papers are devoted to the fabrication of the electrolyte. An all-porous SOFC with a configuration of Ni–YSZ/Ni–YSZ(AFL)/YSZ/LSM–YSZ/LSM was successfully fabricated, where, ingeniously, all the layers were deposited by the electrophoretic method [334]. This yielded a power density of 0.477 W cm$^{-2}$ and OCV of 0.89 V with H$_2$ that reduced to 0.420 W cm$^{-2}$ and 0.8 V in the presence of CH$_4$ as the fuel, the power density, and also the OCV of the cell reduced to, respectively (Figure 24).

For the development of Cu/GDC coating, GDC coating with 20 vol% porosity was obtained by EPD followed by electrodeposition of Cu [335]. The composite coatings were dense and did not require additional heat treatment and possessed high electrical conductivity equivalent to that of Cu metal. Therefore, these materials are promising for use as protective functional coatings for current collectors of SOFC. However, more efforts have to be focused on the scaling-up of this process for industrial-scale applications.

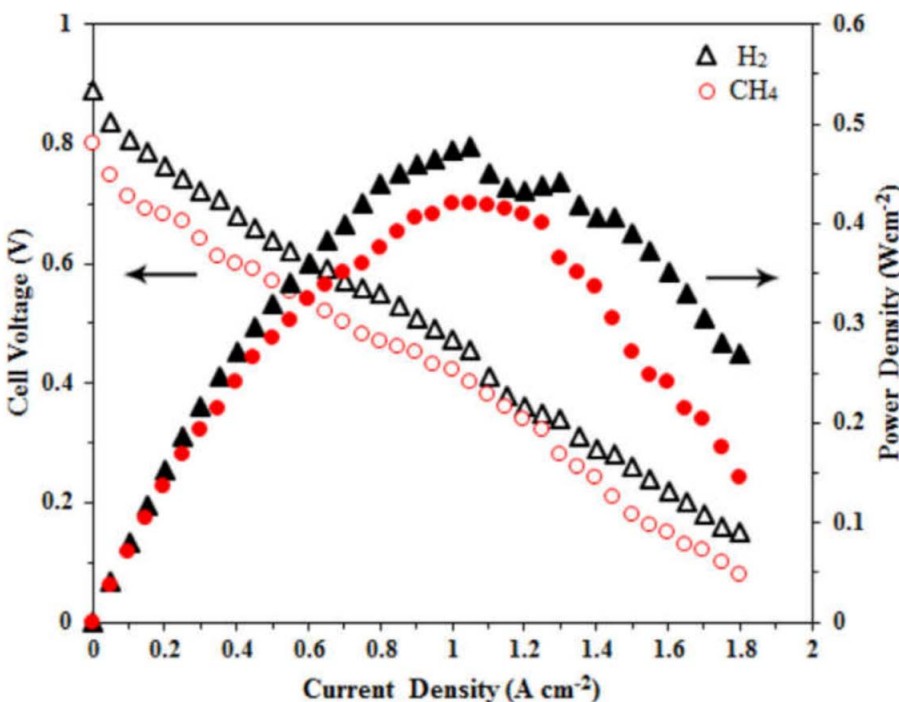

**Figure 24.** Cell voltage and power density curves of the SOFC at 800 °C in $H_2$ and $CH_4$ fuels (adapted from [303]).

8.5.5. Spray Pyrolysis

Spray pyrolysis (SP) deposition is a multi-faceted, cost-effective and industrially scalable method for the synthesis of various compositions of nanoparticles and fabrication of thin films of metals, oxides, etc. [336]. The main advantages of this process over other methods are a smaller number of steps and formation of phase pure coatings. For the SOFC application, the various forms of SP such as conventional spray-pyrolysis deposition, ultrasonic-spray-pyrolysis deposition, electrostatic-spray deposition, and flame-spray deposition (FSD) have been used [337–341]. Composite anode materials with the composition $La_{0.2}Sr_{0.8}TiO_3$–$Ce_{0.8}Sm_{0.2}O_{1.9}$ (LST–CSO) powders prepared from ultrasonic spray pyrolysis were screen-printed on LSGM electrolyte [341]. The cells exhibited a maximum power density of 137.8 mWcm$^{-2}$ at 880 °C, confirming that LST–CSO could be a promising anode for LSGM-based SOFCs [341]. Single perovskites based on Ti-doped $SrFeO_{3-\delta}$ were investigated simultaneously as both anode and cathode for symmetrical SOFCs. Layers with composition $Sr_{0.98}Fe_{1-x}Ti_xO_{3-\delta}$ (x = 0, 0.2, 0.4, and 0.8) were deposited by CSD on YSZ [342]. The electrodes were stable under reducing and oxidizing atmospheres. $Sr_{0.98}Fe_{0.8}Ti_{0.2}O_{3-\delta}$ exhibited Rp values of 0.1 and 0.07 $\Omega$ cm$^2$ in air and $H_2$, respectively, at 700 °C. A 300 μm thick LSGM electrolyte-supported cell generated a power density of 0.7 W cm$^{-2}$ at 800 °C. Though there are many publications on the synthesis of conventional anode materials using the SP method and the fabrication of cells and testing with $H_2$, there are no reports on testing them in the presence of hydrocarbon fuels.

## 9. Nano Anode Composite

It is well documented in the literature that there are beneficial effects of using nano-sized cathode electrodes. Similarly, the benefits of nanocomposite are obvious in anodes, and details are presented below [310].

### 9.1. Synthesis

A representative co-fired Ni–YSZ cermet anode possessing an average nickel particle size of 0.5–1 μm yields TPB densities of 4/μm$^2$, thus offering low anode polarization resistance $R_{P,A} < 0.1$ ohms cm$^2$ at T > 750 °C [310]. The decrease in particle diameter further

increases the TPB length due to the fact that TPB is inversely proportional to the particle diameter. It is understandable from $R_{p,A}$ value that the benefit of reduced particle size of anode composite is marginal compared to the cathode at high temperatures. Nevertheless, the anode resistance increases to extremely larger values for T = 600 °C. Therefore, the benefits of the anode nanocomposite are mostly realized for low-temperature SOFCs. One of the major reasons is believed to be the improved anode kinetics. Nanosize nickel-samaria doped ceria cermet thin films (500 nm) prepared on ScSZ electrolyte supports using reactive radio frequency (RF) sputtering and tested under $H_2$ and a product fuel of $CO_2$ electro-reduced via industrial waste carbon exhibited promising performance [343].

*9.2. Bottlenecks*

Usually, anodes are fabricated by co-firing them at high temperature along with the electrolyte. Generally, electrolytes are sintered at very high temperatures (as high as 1450 °C), which poses unique issues for preventing extensive coarsening/sintering and anode/electrolyte reactions. Hence, the nanocomposite anode creates more intricate issues during fabrication. Another unique problem of the anode is that it contains a metal phase that causes accelerated agglomeration during the typical SOFC operation conditions. The agglomeration of the metallic phase can result in the loss in TPB length [341–344], a loss in percolation [345,346], or mechanical integrity problems whenever the anode is exposed to redox and/or thermal cycles [347,348]. As a consequence of metal phase agglomeration, there will be a drastic increase in the anode's ohmic and/or polarization resistances [340,345].

For instance, several studies were focused on Ni agglomeration [349–354]. The operating temperature and the water vapor content are considered to be the important operating parameters in the steady-state operating conditions of SOFC [349]. The formation of volatile $Ni(OH)_2$ species is deemed to be the primary cause of agglomeration during operation. Additionally, redox-cycling was proved to enhance the Ni agglomeration in Ni-containing anodes [355]. Hence, retention of the anode composite microstructure is more intricate than the fabrication itself.

## 10. Conclusions and Future Perspectives

There are a large number of reports on the use of oxide anodes for testing SOFCs in HC fuel. Among oxide anodes, perovskite manganates and molybdates displayed better performance. Though the oxide anodes do not face any carburization, their catalytic activity failed to match the cermet anode. In fact, the performance of these oxide anodes in HC fuel was half that of $H_2$ fuel. Cu–based cermet systems were targeted due to the carbon retarding ability of Cu atoms. Due to the thermal sensitivity issues, Cu-rich compositions were used only in low-temperature fabrication techniques. Most of these compositions were found to be only suitable for ESC and electrode infiltration techniques. Bimetallic Ni-based compositions showed better thermal stability and are widely used in ASC. Particularly, Ni–Cu compositions exhibit thermal stability as well as resistance to carburization. The commercially available hydrocarbon fuel also contains some $H_2S$ which poisons the catalyst. Ni cermets are prone to sulfidation at high sulfur concentrations. The formation of liquid $Ni_3S_2$ at the operating condition of SOFC would deteriorate the conventional Ni–YSZ anode performance rapidly. However, the Cu or Mo-based cermets are significantly more tolerant to sulfur than the conventional, Ni–based cermets. There were attempts to improve the electrocatalytic activity by different synthesis techniques. Among them, the wet chemical route such as combustion synthesis is found to be impressive. Though nanostructured anode improves the catalytic activity, a practical difficulty is to retain such microstructure over the typical life span (~40,000 h) of SOFC. Still, solution combustion synthesis (SCS) offers an advantage of synthesizing multi-component anode composite with better homogeneity which in turn improves the catalytic activity while using it as an anode functional layer. The present review was devoted to identifying carbon tolerant anode catalyst and its synthesis. Recently, researchers have demonstrated

catalytic partial oxidation as a promising way to control carburization in the anode. Further studies are needed to understand the performance of Ni-alloy catalysts in catalytic partial oxidation conditions.

**Author Contributions:** Writing—original draft, data analysis, and editing; S.S.K., conceptualization and supervision—S.T.A. All authors have read and agreed to the published version of the manuscript.

**Funding:** This research received no external funding.

**Data Availability Statement:** Not applicable.

**Acknowledgments:** The authors thank the Director, CISR-NAL and Head, SED for their encouragement.

**Conflicts of Interest:** The authors declare no conflict of interest.

## Abbreviations

SOFCs—solid oxide fuel cells; HCs—hydrocarbons; HC—hydrocarbon; PEMFCs—proton exchange membrane fuel cells; APU—auxiliary power units; IT-intermediate temperature;YSZ—yttria stabilized cubic zirconia; LSM—strontium doped lanthanum manganite; ASC—anode-supported cells; ESC—electrolyte-supported cells; ScSZ—Scandia stabilized zirconia; DIR—direct internal reforming; ATR—autothermal reforming; POX—partial oxidation; CPOX—Catalytic partial oxidation; TPOX—Thermal partial oxidation; DMO—Direct methane oxidation; OCV—open circuit voltage; TPB—triple-phase boundaries; LSC-lanthanum strontium chromites, LST-lanthanum strontium titanates, LSV—lanthanum strontium vanadates; LC—lanthanum chromite; TEC—thermal expansion coefficient; GDC—gadolinia doped ceria; ASR—area specific resistance; MIEC—mixed ionic electronic conductivity; EPD—electrophoretic deposition; SP—spray pyrolysis.

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
