# Peer review of "Hydrocarbon Compatible SOFC Anode Catalysts and Their Syntheses: A Review"

_2673-4079, doi:10.3390/suschem2040039_

Round 1

Reviewer 1 Report

In this manuscript, anode for direct hydrocarbon oxidation in SOFC are review. In a first part the manuscript different reaction of HCs occurring in the anode of a SOFC are discussed, as well as the problems of carbon deposition in the traditional Ni-YSZ cermets. Mixed perovskite oxides: chromites, manganites, vanadates and layered perovskites are briefly described. In the next section, different cermets are commented: Ni-based, Cu-, Ni-Cu, Fe-Ni, Co-Cu, and Ni-Co. I think that this section of the manuscript is similar to that discussed in the introduction section on drawbacks of Ni-YSZ and could be substantially reduced. After that the problems associated with sulfur poisoning are commented. Finally, different preparation methods are described, such as conventional solid state reaction, wet chemical methods, exsolution of metal nanoparticles, infiltration and physical methods: RF-sputteting. The paper could be of interest for the SOFC community but several revisions are needed as stated below:

  1. 2 should be improved, the order in the photographs is not clear (a) 1. electrolyte preparation, 2 deposition of the anode and then 3 cathode by screen-printing.

(b) 1. Tape-casting electrolyte support (green pellet), 2 lamination electrolyte (white), 3 co-sintering and (4) cathode deposition by screen-printing.

  1. Please, revise the CPOX reaction in Table 1.
  2. Please, check equation (5) on page 7 and (9) on page 8.
  3. The quality of some figures should be improved, for instance Fig. 7.
  4. Revise reference format in all figure captions.
  5. Page 11: “direct” instead of “dire”
  6. Format of Fig. 8.
  7. In Fig. 9. LSCM perovskite is included in other oxides.
  8. The authors only compared the values of total conductivity of the different anode materials; however, area specific resistance are not discussed. I recommend to include a Table to compare the main properties (conductivity, ASR, Power density, etc..) of the most relevant anode materials, which would be useful for the readers.
  9. Page 15: Alternative double perovskite to SMMO have been reported, including Sr2NiMoO6, Sr2NiCoO6, etc.. See for example https://doi.org/10.1016/j.ssi.2013.03.005 and references herein.  
  10. Section 4.1.1.6 cerate zirconates: Ni-BCYYYb is a cermet material. The same comment for Ni-GDC. This section also includes Ni-LSCF and La2NiO4 air electrodes, which decompose in reducing atmosphere. In my opinion, section 4.1.1.6-4.1.2.1 are cermet materials, similar to Ni-YSZ and not mixed oxides.
  11. No conventional anode materials are included: zirconates and uranates. These materials are little studied in the literature to merit a section in a review paper.
  12. 1.3. Other oxides. This section is a little confusing because this includes LSCM, which is previously discussed.
  13. The following sentence needs to be revised on page 17: “peak power density of different oxide anodes reported at 800 °C reported in literature are compared”
  14. 10: “manganites” instead of “manganates”
  15. Ni-YST on page 19.
  16. The order of the different sections should be improved, Ni-YSZ are discussed in the introduction section and then in 4.2.1.
  17. I think that mixed oxides are little discussed compared to cermet materials 4.2.1 to 4.3.6.
  18. The same temperature units should be given in the whole text ºC or K. See figure caption 14.
  19. Alternative synthesis routes could also be briefly commented as spray-pyrolysis, which is useful for large scale production.

Reviewer 2 Report

I have thoroughly read the review article entitled ‘Hydrocarbon Compatible SOFC Anode Catalysts and their Syntheses: A Review’. In my opinion, this work overviews all considerable problems taking place for SOFCs fueled with hydrocarbons. However, there is a number of meaningful comments, which should be considered in detail before its publication in the Sustainable Chemistry journal:

  1. This review is out of novelty, since the most part of references is very old. The authors should perform upgrading references, including recent important works. As can be seen from the PDF-file with my comments, I proposed works for some sections. Other sections need a similar revision.
  2. The authors are invited to check their manuscript again from the viewpoints of terminology, logic structure, and grammar/syntax.
  3. The authors provide various solutions (families of oxide materials, synthesis methods) in many main sections. For each such a section, a comparative analysis should be presented to allow the readers to understand advantages/disadvantages of noted participants of classification. For examples, + and - of different synthesis techniques.
  4. Some additional sections should be presented and discussed. For example, electrophoretic deposition for coatings, long-term stability of hydrocarbon-fueled SOFCs.
  5. Other remarks can be found in the pdf.

Round 2

Reviewer 1 Report

The authors have addressed most of changes proposed by the reviewers; however, some revisions are still needed:

  1. ASR values are reported for numerous mixed oxides.
  2. Manganite is widely used in the literature instead of manganate. Please, check “manganite + SOFC” and “ manganate + SOFC” in Scopus database.

  1. The authors have added the preparation of electrodes by spray-pyrolysis but only cathode materials are discussed. In ref. [324] anode materials, such as nanosize Ni-YSZ, Ni-CGO and mixed oxides, are reviewed and should be included in the manuscript.

This sentence is confusing “Though inventive and efficient SOFC electrode  materials such as La0.6Sr0.4Co0.8Fe0.2O3-δ, Ba0.5Sr0.5Co0.8Fe0.2O3-δ, PrBaCo2O5+δ and Sm0.5Sr0.5CoO3-δ, etc., have been fabricated, they do not have long-term phase stability after operation and are not chemically compatible with the electrolyte [324].” These electrodes were prepared at very low temperature and the chemical and physical compatibility with the electrolyte are improved compared to the traditional screen-printing deposition method due to the lower sintering temperature. Moreover, these electrodes exhibited high stability at the intermediate temperature range”. This issue needs to be addressed.

Reviewer 2 Report

As I can see, all my previous comments have been properly addressed, including growing the existing sections, adding new references. Therefore, this work can be accepted for publication. Along with it, a minor revision in ENG is still required.

Author Response

Thank you for the acceptace.